# Mammary-specific expression of *Trim24* establishes a mouse model of human metaplastic breast cancer

Vrutant V. Shah [1,2,16], Aundrietta D. Duncan [2,3,4,14,16], Shiming Jiang[2,5,16], Sabrina A. Stratton [2,3], Kendra L. Allton [2,6], Clinton Yam[2,7], Abhinav Jain[2,3,4], Patrick M. Krause [1,2], Yue Lu[2,3], Shirong Cai[2,8], Yizheng Tu[2,8], Xinhui Zhou[2,8], Xiaomei Zhang [2,8], Yan Jiang[2,8], Christopher L. Carroll[2,9], Zhijun Kang[2,9], Bin Liu[2,3], Jianjun Shen[2,3], Mihai Gagea [2,10], Sebastian M. Manu[2,3], Lei Huo[2,11], Michael Gilcrease[2,11], Reid T. Powell[12], Lei Guo[12], Clifford Stephan[12], Peter J. Davies[12], Jan Parker-Thornburg [1,2], Guillermina Lozano [1,2,4], Richard R. Behringer[1,2,4], Helen Piwnica-Worms [2,4,8], Jeffrey T. Chang [4,13,17✉], Stacy L. Moulder[2,7,17✉] & Michelle Craig Barton [2,3,4,15,18✉]

Conditional overexpression of histone reader <u>Tri</u>partite <u>m</u>otif containing protein 24 (TRIM24) in mouse mammary epithelia (*Trim24*COE) drives spontaneous development of mammary carcinosarcoma tumors, lacking ER, PR and HER2. Human carcinosarcomas or metaplastic breast cancers (MpBC) are a rare, chemorefractory subclass of triple-negative breast cancers (TNBC). Comparison of *Trim24*COE metaplastic carcinosarcoma morphology, TRIM24 protein levels and a derived *Trim24*COE gene signature reveals strong correlation with human MpBC tumors and MpBC patient-derived xenograft (PDX) models. Global and single-cell tumor profiling reveal *Met* as a direct oncogenic target of TRIM24, leading to aberrant PI3K/mTOR activation. Here, we find that pharmacological inhibition of these pathways in primary *Trim24*COE tumor cells and TRIM24-PROTAC treatment of MpBC TNBC PDX tumorspheres decreased cellular viability, suggesting potential in therapeutically targeting TRIM24 and its regulated pathways in TRIM24-expressing TNBC.

[1] Department of Genetics, The University of Texas MD Anderson Cancer Center, Houston, TX, USA. [2] The University of Texas MD Anderson Cancer Center, Houston, TX, USA. [3] Department of Epigenetics and Molecular Carcinogenesis, Center for Cancer Epigenetics, The University of Texas MD Anderson Cancer Center, Houston, TX, USA. [4] University of Texas MD Anderson Cancer Center UTHealth Graduate School of Biomedical Sciences, University of Texas, Houston, TX, USA. [5] Thoracic Head and Neck Medicine Oncology, The University of Texas MD Anderson Cancer Center, Houston, TX, USA. [6] The Neurodegeneration Consortium, Therapeutics Discovery, The University of Texas MD Anderson Cancer Center, Houston, TX, USA. [7] Breast Medical Oncology, The University of Texas MD Anderson Cancer Center, Houston, TX, USA. [8] Department of Experimental Radiation Oncology, The University of Texas MD Anderson Cancer Center, Houston, TX, USA. [9] Institute of Applied Cancer Science, The University of Texas MD Anderson Cancer Center, Houston, TX, USA. [10] Department of Veterinary Medicine and Surgery, The University of Texas MD Anderson Cancer Center, Houston, TX, USA. [11] Department of Pathology, The University of Texas MD Anderson Cancer Center, Houston, TX, USA. [12] Center for Translational Cancer Research, Institute of Biosciences and Technology, Texas A&M College of Medicine, Houston, TX, USA. [13] Department of Integrative Biology and Pharmacology, University of Texas Health Sciences Center at Houston, Houston, TX, USA. [14] Present address: Salarius Pharmaceuticals, Houston, TX, USA. [15] Present address: Division of Oncological Sciences, Cancer Early Detection Advanced Research, Center Knight Cancer Institute Oregon Health & Science University, Portland, OR, USA. [16] These authors contributed equally: Vrutant V. Shah, Aundrietta D. Duncan, Shiming Jiang. [17] These authors jointly supervised this work: Jeffrey T. Chang, Stacy L. Moulder, Michelle Craig Barton. [18] Lead contact: Michelle Craig Barton. ✉email: jeffrey.t.chang@uth.tmc.edu; smoulder@mdanderson.org; bartonsh@ohsu.edu

Advances in next-generation sequencing (NGS) and mapping of genome-wide epigenetic marks show that epigenetic regulators are frequently mutated and/or aberrantly expressed in numerous diseases, including cancers, often in correlation with poor survival or therapeutic resistance[1]. Among these epigenetic regulators are proteins with structural domains that "read" epigenetic modifications and selectively interact with specific histone or non-histone post-translational modifications (PTMs). These protein PTM readers are critical to interpretation of regulatory information encoded by protein PTMs. The central role of reader domain binding to histone PTMs offers therapeutic opportunities to exploit small molecule ligands that disrupt reader-PTM interactions. As a seminal example, small molecule ligand JQ1 binds the BRD4 bromodomain and disrupts BRD4 interactions with acetylated histone lysines, which leads to inhibition of oncogenic myc-regulated gene networks and potential therapeutic application[2,3].

Our unbiased proteomic screen of embryonic stem cells for endogenous protein interactions with tumor suppressor p53 uncovered Tripartite Motif Protein 24 (TRIM24). TRIM24 is a multi-functional histone reader that ubiquitinates p53, via an N-terminal RING domain, and binds a specific signature of histone PTMs (H3K4me0/H3K23ac) via a combinatorial Plant Homeodomain (PHD) and bromodomain[4–6]. We found that TRIM24 is a co-regulator of estrogen receptor (ER)-regulated genes in breast cancer-derived MCF7 cells[5]. In immortalized human mammary epithelial cells (iHMECs), TRIM24 promotes loss of p53 and metabolic reprogramming that transform iHMECs, which engraft and form high-grade epithelial tumors[7]. High levels of TRIM24, determined by tumor immunohistochemistry (IHC), correlate with poor survival of breast cancer patients[5]. Interestingly, recent reports associate aberrantly high TRIM24 with poor patient prognosis, poor differentiation, advanced stage, chemo-resistance, and recurrence in a wide range of cancers, especially solid tumors[8–10].

Given the functional studies in cellulo and correlative links between TRIM24 and human cancers, we hypothesized that TRIM24 is a potent oncogene that reprograms transcriptional networks and promotes development of aggressive breast cancers. We tested our hypothesis in a physiologically relevant setting by piggyBac transposon-mediated integration of a single copy of Trim24 under conditional expression control in the mouse genome. Here, we show that <3-fold overexpression of Trim24 in mammary epithelia is sufficient to induce mammary tumors, 67% of which are characterized as carcinosarcomas.

In breast cancer patients, carcinosarcomas are included within the diagnosis of metaplastic breast cancers (MpBC), a rare subgroup of breast cancers that are commonly triple negative (TNBC) and diagnosed by histological evidence of at least two cell types within an individual tumor: an epithelial component (carcinoma) and a non-epithelial component (sarcoma, matrix producing or squamous morphology)[11]. MpBCs are chemo-refractory and have a worse prognosis compared to non-metaplastic breast cancers, though little is known about the origin or drivers of these tumors[11]. Markers of epithelial–mesenchymal transition (EMT) are increased in expression in MpBC, compared to other subtypes of breast cancer[12]. MpBCs are usually high grade with larger tumor size, though often lymph node negative, at the time of presentation[13]. MpBCs also have frequent aberrations in TP53 and PI3K/AKT/ mammalian target of rapamycin (mTOR) pathways, as well as prominent expression of vascular endothelial growth factor (VEGF) and programmed cell death-ligand 1 (PD-L1)[14,15]. Molecular characterization by next-generation sequencing (NGS) of MpBC patient tumors and patient-derived xenografts (PDXs) of MpBC revealed some potential targets; however, there is limited ability to define a molecular landscape with a large patient cohort in this rare TNBC subtype and few animal models exist to test novel targeted regimens[14,16–18].

Here, we present evidence that TRIM24 is an oncogenic histone reader with the ability to drive an aggressive breast cancer phenotype in a mouse model with pathologic and molecular features that parallel MpBC human patient data. Our data suggest that TRIM24 may serve as a prognostic biomarker, as well as a potential predictor of disease response to select targeted therapies. Global gene expression analyses of TRIM24-driven carcinosarcomas were used to generate a TRIM24 metaplastic signature that revealed highly up-regulated glycolysis, EMT and PI3K pathways, the latter confirmed by global and single-cell analysis of tumor protein levels. TRIM24 aberrantly activates c-MET-PI3K-mTOR pathways, nominating pharmacological approaches that proved effective in inhibition of metaplastic carcinosarcoma primary cell viability. We assessed degradation of TRIM24, as a potential therapeutic approach, using MpBC PDX models that express high levels of TRIM24. A recently developed proteolysis-targeting chimera (PROTAC) fusion of small molecule IACS-9571, a bromodomain inhibitor of TRIM24, and the Von-Hippel Lindaeu (VHL) ligand[19,20] inhibited viability of TRIM24-expressing MpBC and non-MpBC TNBC PDX tumorspheres. Thus, a mouse model of human MpBC, driven by histone reader TRIM24, was used to nominate PI3K/mTOR inhibitors and a TRIM24-specific PROTAC for further assessment as potential therapeutic approaches for a rare subclass of chemorefractory breast cancer patients.

## Results

**TRIM24 over-expression drives mammary gland tumorigenesis.** To conditionally express Flag-tagged, murine Trim24 in mammary epithelia, MMTV-driven Cre mice were bred with mice bearing a floxed ß-Geo-Stop-mTrim24-flag cassette transgene. Trim24$^{LSL}$;MMTV-Cre$^{Tg/0}$, hereafter referred to as Trim24$^{COE}$, were viable, fertile, and over-produced TRIM24 in mammary epithelia (Supplementary Fig. S1a–d). The Flag-tagged mTrim24 transposon integrated as a single copy, into the q-arm of chromosome 1 without disrupting any known genes or regulatory elements, as determined by whole-genome sequencing and PCR (Supplementary Fig. S1e and Supplementary Data 3). Trim24 transcript expression in the whole mammary gland was increased ~2.5–3-fold in Trim24$^{COE}$ mice, as compared to controls (Supplementary Fig. S1c). TRIM24 and exogenous Flag-TRIM24 protein levels in the mammary epithelium were confirmed by western blot analysis (Supplementary Fig. S1d). Assessment of an aging cohort of virgin Trim24$^{COE}$ females (n = 40) revealed atypical hyperplasia of mammary ductal and lobular epithelia, ductal thickening, as well as increases in both side branching and terminal endbuds within 2 months (Fig. 1a, b). Trim24$^{COE}$ mice developed palpable mammary tumors within 5–14 months of age (penetrance of 46%). Necropsy revealed no other primary tumor sites and two mice with metastases: one to the lung and the other to the liver (Supplementary Data 4). Trim24$^{LSL}$, lacking MMTV-Cre-expression, lacked any palpable tumor formation within 600 days and MMTV-Cre-only mice (n = 27) had occasional hyperplasia and spontaneous mammary tumor development (Fig. 1c). Trim24$^{COE}$ tumors overexpressed TRIM24 compared to controls, as determined by IHC (Fig. 1d). Trim24$^{COE}$ tumors exhibited overexpression of FLAG and TRIM24 proteins and lacked detectable expression of p53 (Fig. 1e), compared to Cre-only control mammary glands. These results show that Trim24$^{COE}$ mice develop mammary tumors at high penetrance, and these tumors maintain expression of Flag-tagged TRIM24.

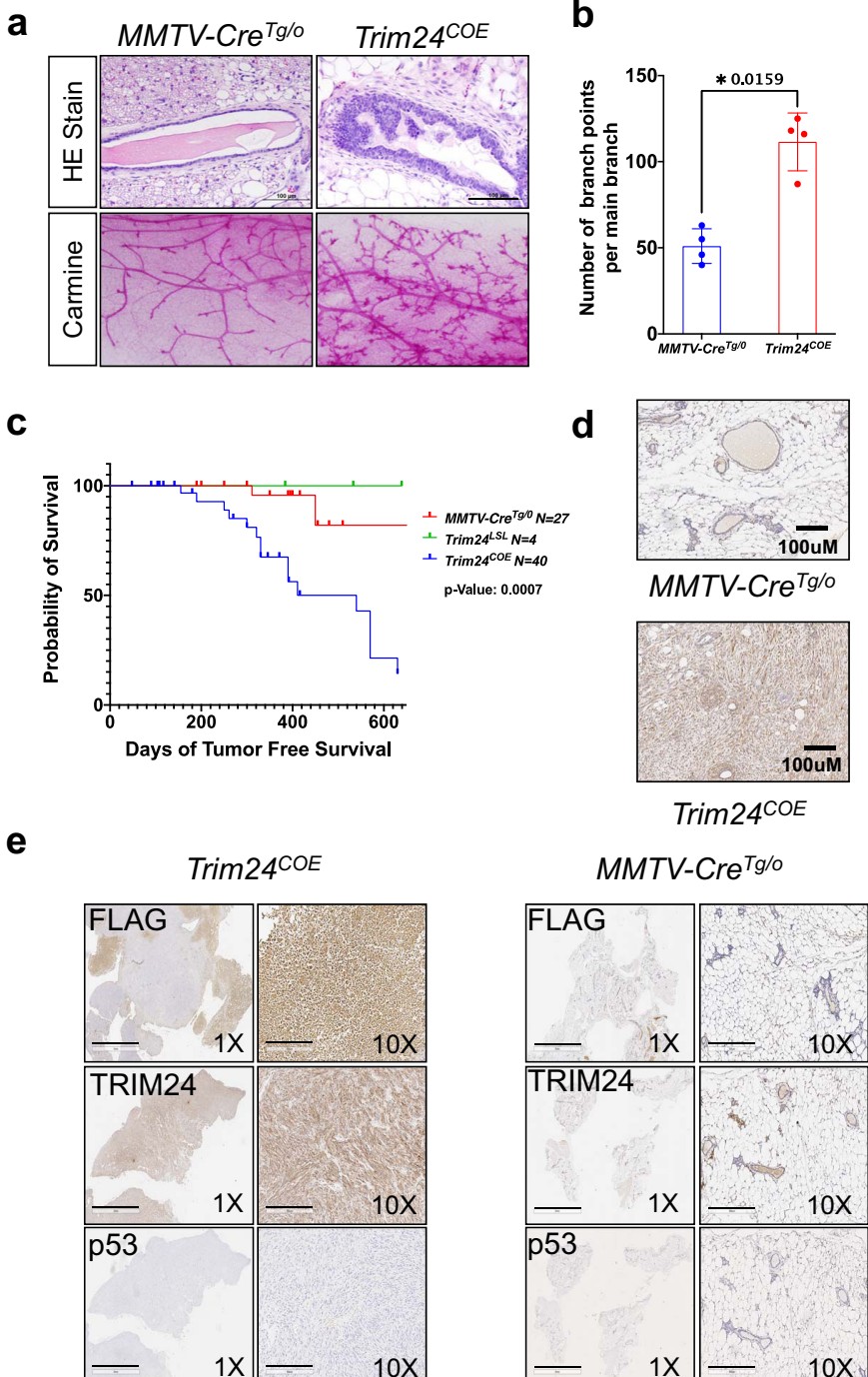

**Fig. 1 TRIM24 over-expression promotes hyperplasia, increases ductal branching, and tumorigenesis. a** Top panels show H&E staining and bottom panels carmine alum staining of mammary glands of 2-month-old *Trim24*COE and control *MMTV-Cre*Tg/0. **b** Quantified comparison of mammary gland branch points per ductal main branch of 2-month old (*n* = 4 mice each) *MMTV-Cre*Tg/0 (blue) and *Trim24*COE (red) mice. Data represented as mean with SD and *p*-value (*<0.05) is calculated using two-tailed paired t test. **c** Tumor-free percent survival curve of *MMTV-Cre*Tg/0, *Trim24*LSL, and, *Trim24*COE mice. The *p*-value is calculated based on Log-rank method. **d** Comparison of TRIM24 IHC and morphology of age-matched *MMTV-Cre*Tg/0 normal mammary gland and *Trim24*COE tumor sections. Images are representative of >50 experiments. Scale bar 100 μM. **e** IHC of *Trim24*COE tumor sections (left) and mammary glands of *MMTV-Cre*Tg/o mice (right) with antibodies recognizing FLAG, TRIM24, and p53 at two magnifications: ×1 and ×10. Images are representative of 15 experiments. Scale bar = 3 mM (1×) and 300 μM (10×).

**A majority of TRIM24 over-expressing tumors are carcinosarcomas**. Pathologic examination of mammary tumors formed in *Trim24*COE mice revealed three tumor types, including adenoma (8%), carcinoma (25%), and carcinosarcoma (67%) (Fig. 2a and Supplementary Data 4). Carcinosarcoma or metaplastic tumors are composed of two different tissue types: carcinoma (cancer of epithelial cells) and sarcoma (cancer of connective tissues or mesenchymal cells). A majority of human carcinosarcoma or metaplastic breast cancers (MpBC) are classified broadly as triple-negative breast cancer (TNBC) or claudin low and are highly associated with two major hallmarks of EMT, a loss of E-cadherin and gain of vimentin[12]. Assessment of

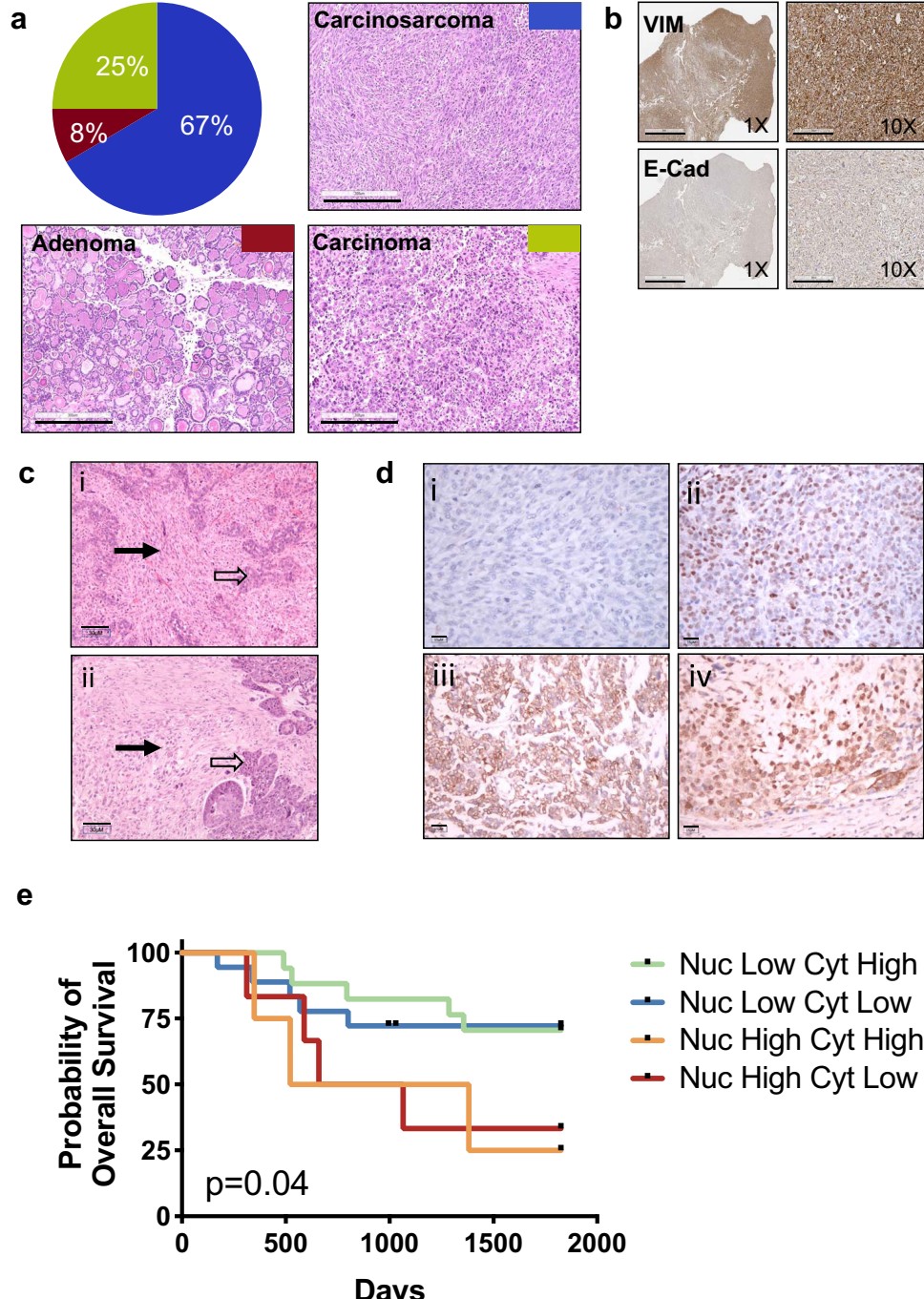

**Fig. 2 TRIM24 over-expression promotes development of mammary metaplastic carcinosarcomas with similarity to tumors borne by MpBC patients with poor probability of survival. a** Pie chart showing quantified classifications (% of total) of *Trim24*^COE mammary tumors along with representative H&E stained sections of *Trim24*^COE mammary tumors: metaplastic carcinosarcomas (blue); carcinoma (green); and adenoma (red). Images are representative of 15 experiments. Scale bar = 300 μM. **b** Representative IHC of metaplastic carcinosarcomas: vimentin and E-Cadherin at ×1 and ×10 magnification (left panel and right panel, respectively). Images are representative of >25 experiments. Scale bar = 3 mM (1×) and 300 μM (10×). **c** Comparison of H&E stained sections of (i) murine *Trim24*^COE tumors and (ii) human MpBC tumor. Each tumor has a cohesive epithelial component (open arrow) and a discohesive spindle cell component (solid arrow). Images are representative of >10 experiments. Scale bar = 30 μM **d** Representative IHC staining of TRIM24 in human MpBC biopsies, illustrating (i) negative nuclear and cytoplasmic staining; (ii) positive nuclear staining and negative cytoplasmic staining; (iii) negative nuclear staining and positive cytoplasmic staining; and (iv) positive nuclear and cytoplasmic staining (immunoperoxidase with 3,3'-diaminobenzidine chromogen). Images are representative of 47 experiments. Scale bar = 15 μM **e** Kaplan–Meier plot of probability of overall survival versus time (days) for MpBC patient (n = 47) shows nuclear TRIM24 is linked to poor MpBC patient survival regardless of cytoplasmic TRIM24 protein levels. Each line in the graph represents survival data for MpBC patients segregated by pathologist-scored TRIM24 nuclear (Nuc) and cytoplasmic (Cyt) levels.

$Trim24^{COE}$ carcinosarcoma tumors by IHC revealed gain of vimentin, loss of E-cadherin, estrogen receptor (ER) and progesterone receptor (PR), and no over-expression of receptor tyrosine kinase erbB-2 (ERBB2) (Fig. 2b and Supplementary Fig. 2). Thus, like a majority of human MpBC, $Trim24^{COE}$ carcinosarcomas are triple-negative.

Human MpBC is diagnosed by pathologists via light microscopy to determine the presence of carcinoid and sarcomatoid regions in sectioned tumor samples[21]. Pathologist evaluation of H&E stained, formalin-fixed, and paraffin-embedded (FFPE) sections of $Trim24^{COE}$ mouse carcinosarcomas (Fig. 2c) revealed considerable, morphological similarities between human metaplastic breast and $Trim24^{COE}$ tumors with both sarcomatoid and carcinoid regions. Expression of TRIM24 was additionally assessed by IHC of a tumor array with 46 MpBC patient biopsies and quantified according to previously published convention (Supplementary Data 5)[22]. This revealed four different expression patterns: (i) low staining of TRIM24 in the nucleus and cytoplasm, (ii) high nuclear TRIM24 expression and little cytoplasmic staining, (iii) high TRIM24 expression in the cytoplasm and little or no TRIM24 in the nucleus and, (iv) high TRIM24 expression in both nucleus and cytoplasm (Fig. 2d). Survival curves were generated with patients divided into classes i–iv by scored TRIM24 levels. MpBC patients were morphologically classified as subtypes: spindle cell (32%), matrix producing (43%), mixed spindle cell/matrix producing (15%), chondrosarcomatoid (2%), osteochondroid (2%), and squamous (2%). High nuclear TRIM24 levels were found in 27% spindle and 45% matrix producing subtyped MpBC patients, similar to the cohort distribution of subtypes (Supplementary Data 5). Interestingly, high nuclear TRIM24, and not associated cytoplasmic TRIM24, levels correlate with worse 5-year overall survival ($p = 0.04$) (Fig. 2e). Thus, nuclear functions of TRIM24 in chromatin association and coregulation of transcription may have greater oncogenic significance than more recently reported cytoplasmic functions of TRIM24 in antiviral immunity[23].

**TRIM24 up-regulates EMT and glycolysis in mammary tumors.** TRIM24 is a histone "reader" protein, which impacts transcription of a number of genes involved in metabolism and nuclear receptor-regulated functions, as we previously showed by Nanostring analyses of genes differentially regulated in response to ectopic expression *in cellulo*[5,7]. To determine the regulatory impact of TRIM24-driven mammary tumorigenesis in a physiologically relevant mouse model, we performed deep sequencing of RNA (RNA-seq) isolated from $Trim24^{COE}$ tumors ($n = 6$, 3 metaplastic carcinosarcoma and 3 carcinomas), control un-diseased mammary glands from age-matched MMTV-Cre animals ($n = 4$) and mammary tumors spontaneously generated in either FVB or FVB/129 mice in a non-TRIM24 overexpressing background ($n = 3$). We plotted the overall distribution of differentially expressed genes (DEGs), using a volcano plot to illustrate up- and down-regulated genes, significantly altered by TRIM24 over-expression (Fig. 3a). Expression of 2221 of these genes was up-regulated, whereas 2826 were down-regulated. RNA-sequencing data were analyzed using DESeq2 to determine DEGs with TRIM24 over-expression[24]. Using a false discovery rate (FDR) of 5%, we found 2006 genes, 529 up-regulated and 1477 down-regulated, had at least 5-fold change (Supplementary Data 6). An unbiased clustering analysis showed a clear distinction between gene expression profiles of MMTV-Cre-expressing, normal control mammary glands, non-TRIM24-overexpressing mammary tumors and TRIM24-overexpressing mammary tumors. Metaplastic carcinosarcomas clustered separately from carcinomas; considerable heterogeneity occurs most notably among carcinomas (Fig. 3b).

To validate the expression of genes within the top 1% (based on FDR) of DEGs in our RNA-Seq analysis, we used qRT-PCR analysis of RNA isolated from $Trim24^{COE}$ tumors (carcinoma, $n = 3$: tumor number 2735, 3956, and 89; metaplastic carcinosarcoma, $n = 3$: tumor number 567, 64, and 897) and MMTV-cre controls (Supplementary Fig. 3a). As predicted by RNA-seq, qRT-PCR shows that Trim24, gelactin-1 (Lgals1), macrophage migration inhibitory factor (Mif), cyclin-dependent kinase Inhibitor 2 A (Cdkn2a), vimentin (Vim), and fibronectin 1 (Fn1) were highly expressed in TRIM24-driven tumors compared to Cre-only mammary glands (Supplementary Fig. 3b). In comparison, p53 (Trp53), aldehyde dehydrogenase 1 family member A1 (Aldh1a1), V-set domain-containing T-cell activation inhibitor 1 (Vtcn1), synuclein gamma (Sncg), leukocyte receptor tyrosine kinase (Ltk), E-cadherin (Cdh1), and synaptotagmin 9 (Syt9) were down-regulated in TRIM24-driven tumors compared to Cre-controls (Supplementary Fig. 3a, b). Consistent with high EMT association with MpBC, we saw that differential expression of EMT markers, for example, up-regulated Vim and Lgals1 and down-regulated Cdh1, was greater in metaplastic carcinosarcomas compared to carcinomas, consistent with protein levels of E-cadherin and vimentin, identified via IHC in TRIM24-driven tumors (Fig. 2b). Together, these results show that TRIM24-driven tumors express a distinct set of genes that can distinguish between metaplastic and carcinoma tumors.

To determine the functional significance of TRIM24-impacted gene regulation in mammary metaplastic carcinosarcoma tumors, we identified hallmark pathways of differentially expressed genes using the ssGSEA algorithm[25]. We assessed differential pathway activity using a Student's t-test, correcting for multiple hypotheses and accepting pathways with a 5% FDR cutoff. Although the molecular profiles of transcription showed some heterogeneity between tumor samples of the same pathological classification (Fig. 3b), we identified several significant hallmark pathways of $Trim24^{COE}$ metaplastic carcinosarcomas (Fig. 3c). The top five, significantly up-regulated pathways in TRIM24-driven tumors were the glycolysis pathway, EMT, angiogenesis, allograft rejection, and mTORC1 signaling (Fig. 3c). The only significant pathway down-regulated by TRIM24 expression is a group of genes (KRas signaling DN) likewise down-regulated in response to oncogenic KRAS expression[26]. The predicted scores of statistically significant pathways (>5% FDR) were clustered, revealing that metaplastic carcinosarcoma tumors (M, $n = 3$) had similar scores that were significantly different from normal tissue[27]. In contrast, the scores for carcinomas (C, $n = 3$) were heterogeneous and in between the scores of metaplastic carcinosarcoma tumors (Fig. 3d)[28]. Additionally, EMT scores, calculated by an ssGSEA algorithm, showed significant differences between TRIM24-driven metaplastic carcinosarcoma tumors and TRIM24-driven carcinoma tumors. Interestingly, the EMT scores of TRIM24-driven and non-TRIM24-driven carcinomas were not significantly different (Supplementary Fig. 3c).

To investigate the association between TRIM24 activity and metaplastic tumors, we analyzed previously published tumor expression data sets. Having shown previously[5,6] that TRIM24 is regulated by post-transcriptional mechanisms, we used a $Trim24^{COE}$ expression signature of downstream marker genes to assess TRIM24 activity levels, rather than evaluating TRIM24 gene expression alone. Weigelt et al.[29] determined gene expression and copy number variation of 26 human MpBC patients and reported specific features that correlate with histological subtypes. We compared the gene expression analysis of Weiget et al. by means of the calculated TRIM24 metaplastic score and found that this cohort of human MpBC tumors has a significantly higher TRIM24 metaplastic score than TRIM24-driven carcinomas, as did TRIM24-driven metaplastic carcinosarcomas

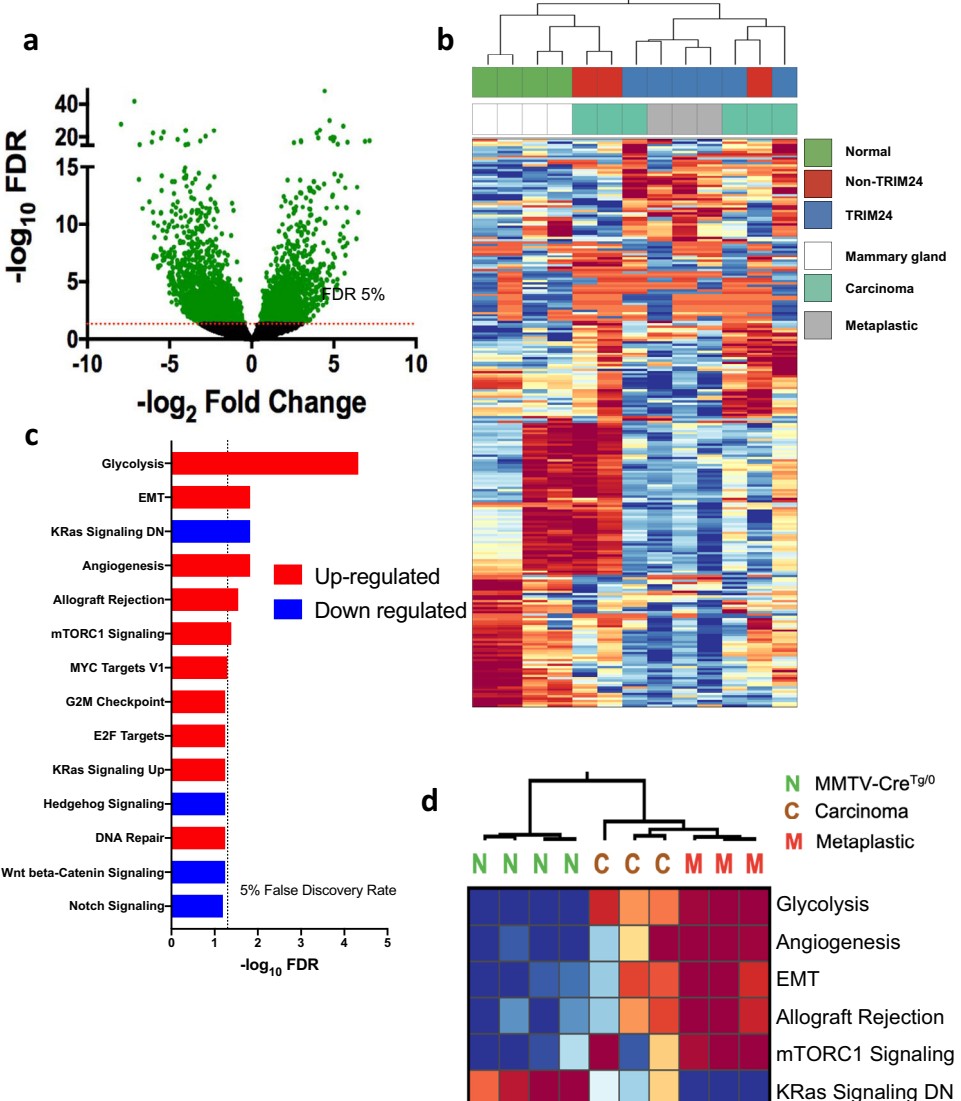

**Fig. 3 TRIM24 over-expression drives expression of EMT-related genes. a** Volcano plot of all genes. Green dots represent all genes which exceeded FDR cutoff. Y-axis represents $-\log_{10}$ of FDR and X-axis represents $-\log_2$ fold change; genes up-regulated on the right and genes down-regulated on the left. Dashed line represents FDR < 0.05 threshold **b** Unsupervised hierarchical clustering identifies three distinct groups from RNA deep sequencing. The heatmap is prepared using diverging scale where up-regulated genes are in red and down-regulated genes are in blue. The first group represents *MMTV-Cre^{Tg/0}* mammary glands (green), second cluster is *Trim24^{COE}* driven mammary tumors (both carcinomas and metaplastic carcinosarcomas), and the third group is mixed non-TRIM24 driven mammary tumors and *Trim24^{COE}* carcinomas. These clusters are further classified into normal *MMTV-Cre^{Tg/0}* mammary glands (white), carcinomas (teal), and metaplastic carcinosarcomas (gray). **c** Gene set enrichment analysis of up-regulated (red) and down-regulated (blue) differentially expressed pathways between *Trim24^{COE}* metaplastic carcinosarcoma tumors compared to *MMTV-Cre^{Tg/0}* mammary glands. The dotted line represents FDR < 0.05 threshold. **d** Heatmap of top differentially expressed pathways calculated by ssGSEA scores between N - *MMTV-Cre^{Tg/0}* mammary glands, C - *Trim24^{COE}* carcinomas and M - *Trim24^{COE}* metaplastic carcinosarcoma tumors.

(Supplementary Fig. 3d), predicting that TRIM24 is activated in human metaplastic tumors. Previously a mouse model with MMTV-driven expression of Hepatocyte Growth Factor Receptor *Met* tyrosine kinase was reported to develop diverse EMT-high and claudin-low, metaplastic carcinosarcoma tumors[27,30]. Comparison of our calculated TRIM24 metaplastic score to microarray expression analysis of MMTV-Met-driven and MMTV-Met;Trp53^{fl/+} tumors revealed that the TRIM24 signature is significantly higher in the two MET-driven metaplastic models, but not the non-metaplastic p53/MET model (Supplementary Fig. 3e). Next, the association between TRIM24-driven metaplastic carcinosarcoma tumors and a previously reported human MpBC gene expression signature was calculated using GSEA[12]. This assessment revealed significant alignment between *Trim24^{COE}*

carcinosarcoma DEGs and human metaplastic gene signature data (NES = 1.615 and FDR 0.002) (Supplementary Fig. 3f). Together these expression analyses support our morphological and molecular characterization of mammary-specific *Trim24^{COE}* tumors as a model of MpBC.

**PI3K pathways are up-regulated in TRIM24-driven mammary tumors.** We used Reverse Phase Protein Array (RPPA) to quantify specific proteins in TRIM24-driven metaplastic carcinosarcomas compared to *MMTV-Cre* mammary glands. Two biological replicates of *Trim24^{COE}* metaplastic carcinosarcomas with two technical replicates were compared to *MMTV-Cre* mammary glands. There were 247 protein probes used to determine protein levels and all relative protein levels were normalized

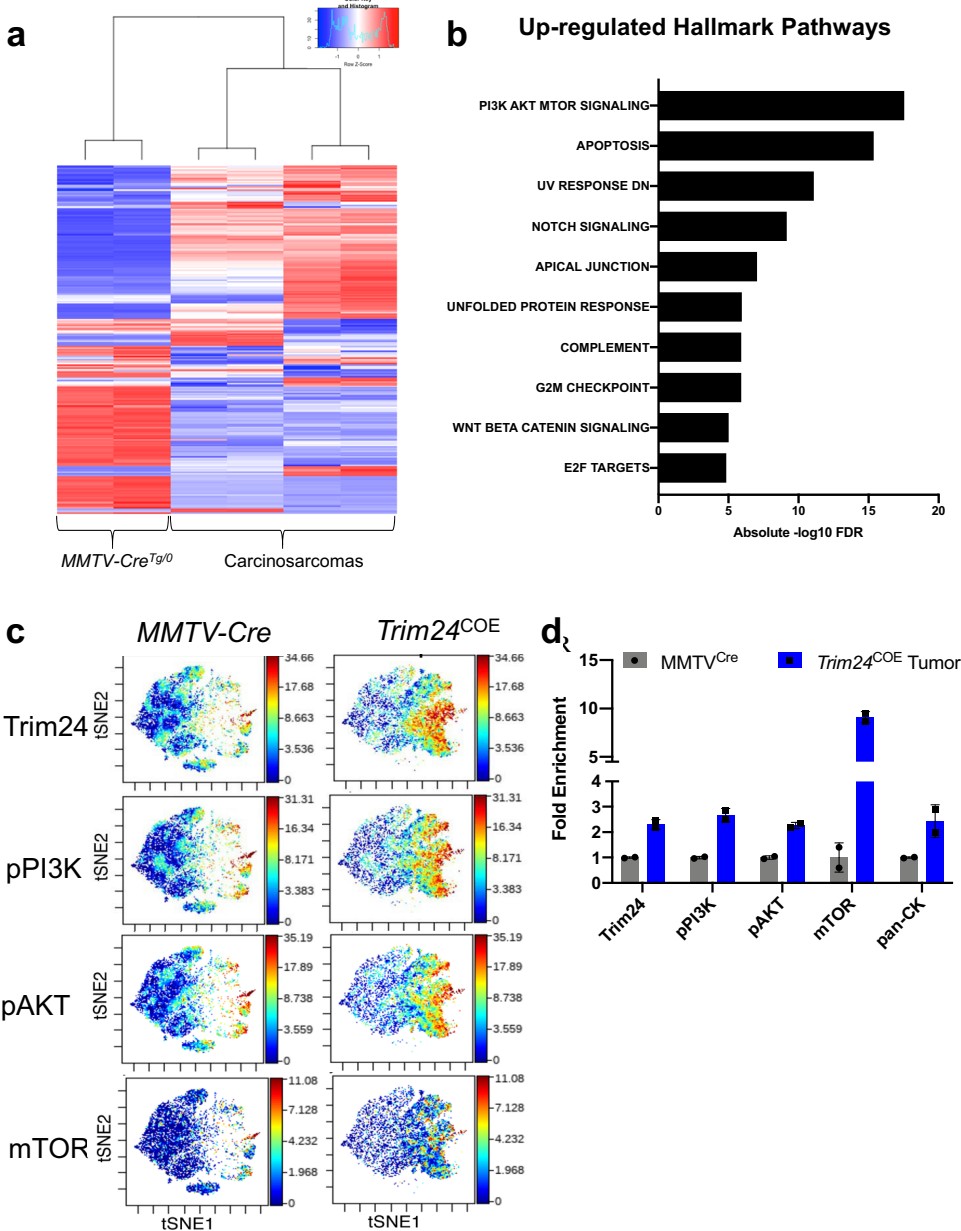

**Fig. 4 Protein profiling shows TRIM24 over-expression correlates with increased PI3K-AKT-mTOR signaling. a** Reverse Phase Protein Array shows distinct protein profiles between metaplastic carcinosarcoma tumors driven by TRIM24 compared to *MMTV-Cre^{Tg/0}* mammary glands shown by heatmap. **b** The bar plot represents up-regulated hallmark pathways on proteins identified by RPPA. X-axis represents $-\log_{10}$ FDR values for each hallmark pathways. **c** Validation of protein expressions by CyTOF where allograft tumors derived of TRIM24 overexpressing metaplastic carcinosarcoma primary cell lines where compared to *MMTV-Cre^{Tg/0}* mammary glands for TRIM24, phospho-PI3K, phospho-AKT, and mTOR. **d** Quantification of overall median protein expression in *Trim24^{COE}* allograft tumors compared to *MMTV-Cre^{Tg/0}* mammary glands ($n = 2$ biologically independent samples). Data represented as mean with SD.

using protein loading controls[31]. Normalized protein loading and transformed linear values were used to determine fold change and later log 2 transformed fold change values (Supplementary Data 7). The normalized values were also used to generate a heatmap representation of differentially produced (57 up-regulated and 47 down-regulated) proteins in *Trim24^{COE}* tumors compared to control mammary glands (Fig. 4a). We converted protein probes to gene identifiers (IDs) to perform GSEA and determine hallmark pathways of differentially expressed proteins[25]. Based on RPPA, we identified PI3K-mTOR signaling, apoptosis, UV response and Notch signaling as the top up-regulated pathways (Fig. 4b). mTOR complex pathway and G2/M

check point genes were both present in up-regulated hallmark pathways determined by RNA-Seq and RPPA (Figs. 3c, d and 4b).

To assess mTOR signaling in *Trim24^{COE}* metaplastic carcino-sarcomas at a single-cell level, we performed cytometry by time-of-flight (CyTOF) using allografted tumors. These allografted tumors were generated by injecting mouse primary cell lines, derived from *Trim24^{COE}* metaplastic carcinosarcoma tumors, into mammary fat pads of nude mice[32]. CyTOF analysis revealed that *Trim24^{COE}* metaplastic carcinosarcomas exhibited high levels of TRIM24, phospho-PI3K (pPI3K), phospho-AKT (pAKT), and mTOR within the same spatial orientation (Fig. 4c). These results were quantified using an overall median of protein levels in

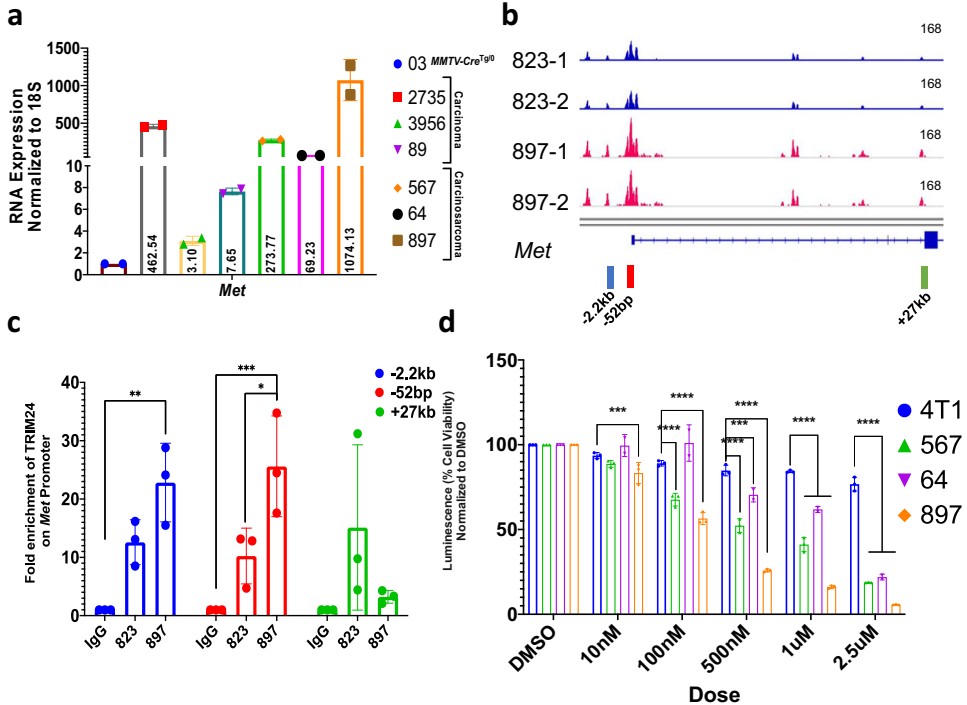

**Fig. 5 TRIM24 directly activates c-Met to support metaplastic carcinosarcoma cell viability. a** qRT-PCR validation of cMET RNA expression in TRIM24 overexpressing tumors compared to *MMTV-Cre*$^{Tg/0}$ mammary gland (03) (*n* = 2 technical replicates for each tumor samples). **b** Snapshot of Integrated Genome Viewer (IGV) showing tracks and peaks of ATAC-Seq-determined chromatin accessibility of *MMTV-Cre* spontaneous tumor cell line 823 (blue) and TRIM24 overexpressing primary metaplastic carcinosarcoma cell line 897 (red). The scales of the tracks are adjusted to 168. The color bar below the tracks shows positions of primers used to assess TRIM24 interactions with cMET chromatin. **c** Chromatin immunoprecipitation (ChIP) PCR showing enrichment of TRIM24 on *Met* promoter to validate ATAC-Seq on 823 and 897 cell lines. The color of each bars indicates primers corresponding to arrows shown in B (*n* = 3 biological replicates). (*\*p* = 0.0209, *\*\*p* = 0.0014, *\*\*\*p* = 0.0004). **d** Cell viability assay on 4T1 (mouse TNBC cell line- non-metaplastic) and TRIM24 overexpressing primary cell lines (567, 64, and 897) upon treatment with Crizotinib, an FDA-approved ATP-competitive selective small molecule inhibitor of c-Met and ALK receptor. Percent cell viability is measured using a luminescence reader; values are normalized to DMSO, a vehicle control (*n* = 3 biological replicates). Data represented as mean with SD in **a**, and mean ± SEM in **c**, **d**. *p*-values are calculated in **c** based on two-way ANOVA for multiple comparisons using Tukey method and in **d** using two-way ANOVA for multiple comparisons using Holm–Sidak method. (*\*\*\*p* < 0.001, *\*\*\*\*p* < 0.0001).

*Trim24*$^{COE}$ metaplastic carcinosarcomas compared to MMTV-Cre mammary glands along with pan cytokeratin to differentiate sarcomatoid carcinoma from sarcoma (Fig. 4d). Thus, CyTOF supports the conclusion that overexpression of TRIM24 up-regulates key members of the PI3K-AKT-mTOR pathway, leading to aberrant activation frequently associated with tumorigenesis, including MpBC in human patients[14].

**TRIM24 activates c-Met and PI3K pathway in carcinosarcomas.** Activation of the PI3K-AKT-mTOR pathway in TRIM24-driven carcinosarcoma tumors nominates potential therapeutic targets[14]. In particular, both RNA-seq and qRT-PCR revealed that *Met*, an upstream regulator of the PI3K-AKT-mTOR pathway, is overexpressed (RNA-Seq log$_2$ fold = 3.9) in TRIM24-driven tumors, compared to Cre-only mammary glands (Fig. 5a). To determine a mechanism for TRIM24-mediated activation of *Met* expression (c-Met), previously reported to drive formation of mammary metaplastic carcinosarcoma tumors when overexpressed[27,30], we performed Assay for Transposase Accessible Chromatin with high-throughput sequencing (ATAC-Seq) of murine primary cell lines; an *MMTV-Cre*-only cell line (823), derived from a spontaneous mammary outgrowth, and a *TRIM24*$^{COE}$ metaplastic carcinosarcoma-derived primary line (897). The TRIM24-driven metaplastic carcinosarcoma cell line (897) showed higher chromatin accessibility at specific sites along

the *Met* gene region, compared to the *MMTV-Cre*-control cell line (823) (Fig. 5b). We used chromatin immunoprecipitation (ChIP) PCR analysis of primary tumor cell lines and confirmed that TRIM24 was significantly enriched at regions of increased chromatin accessibility near −2.0 kb and the proximal promoter (−50 bp) of *Met* in the primary metaplastic carcinosarcoma line (897) but not the *MMTV-Cre* control (823) (Fig. 5c). Further, using a tet-inducible system, we showed that knock-down of *Trim24* leads to a parallel decrease in mRNA expression of *cMet* (Supplementary Fig. 4a). Enrichment of TRIM24, as a co-regulator of transcription that recognizes a dual signature of acetylated and unmethylated histone H3 (H3K23ac:H3K4me0) through its combinatorial PHD/bromodomain[5], offers a mechanism for increased chromatin accessibility, activated transcription and high levels of *Met* gene expression revealed by RNA-seq (Fig. 5a).

We leveraged our finding that TRIM24 directly activates *Met* expression and *TRIM24*$^{COE}$ tumor-derived primary cells to test the potential of therapeutically targeting MET. Crizotinib is a dual inhibitor of MET and anaplastic lymphoma kinase (ALK) that has shown promise in multiple cancer types, including breast cancer[33–35]. Crizotinib is an FDA-approved drug for metastatic non-small cell lung cancer (NSLC) and anaplastic large cell lymphoma (ALCL), which has been used in more than 70 clinical trials in United States (clinicaltrials.gov) for various cancers[36]. Crizotinib treatment of *TRIM24*$^{COE}$ primary carcinosarcoma cell

lines (567, 64, and 897), compared to a murine TNBC-like cell line 4T1, led to a significant, $TRIM24^{COE}$ metaplastic carcinosarcoma cell-specific reduction in cell viability in a dose-dependent manner (Fig. 5d). To further assess MET as a drug target, we used a selective inhibitor of MET kinase, PHA-665752, previously shown to reduce tumor burden in a mouse model of gastric cancers where *MET* is amplified[37]. PHA-665752 treatment and targeting of MET significantly reduced cell viability in a dose-dependent and metaplastic carcinosarcoma-specific manner (Supplementary Fig. 4b).

Aberrant activity of MET potentially disrupts cellular homeostasis by activation of one or more downstream pathways, including PI3K[38]. PI3K pathway components exhibited high protein levels in RPPA and CyTOF analysis of $TRIM24^{COE}$ metaplastic carcinosarcoma tumors (Fig. 4), likely activated by post-transcriptional regulation (Supplementary Fig. 4c). To focus downstream of MET and determine whether the PI3K pathway plays an essential role in $TRIM24^{COE}$ metaplastic carcinosarcomas, we used Dactolisib, a dual inhibitor of PI3K isoforms and mTOR[39,40]. We found that $TRIM24^{COE}$ metaplastic carcinosarcoma cell lines showed a marked reduction in cell viability with low doses of Dactolisib, which was not recapitulated in the 4T1 TNBC-like cell line (Supplementary Fig. 4d).

Together, we show that TRIM24-mediated, epigenetic activation of c-Met is disrupted by selective inhibitors, Crizotinib and PHA-665752, in a dose-dependent manner in primary metaplastic carcinosarcoma tumor cells. c-Met induces up-regulation of PI3K and mTOR pathways at a protein level, consistent with our RPPA and CyTOF analysis of $TRIM24^{COE}$ metaplastic carcinosarcomas. Using a dual inhibitor developed for human PI3K isoforms and mTOR, we showed that viability of murine $TRIM24^{COE}$ metaplastic carcinosarcoma cells was likewise inhibited, further supporting the parallels between human cancers and tumors arising in the $TRIM24^{COE}$ mouse model.

**TRIM24 metaplastic carcinosarcomas are similar to human disease.** To compare $Trim24^{COE}$ metaplastic carcinosarcoma mammary tumors directly to human metaplastic breast cancer (MpBC) samples, we analyzed patient data generated by an ongoing clinical trial: A Robust TNBC Evaluation fraMework to Improve Survival (ARTEMIS: MD Anderson protocol 2014-0185; NCT02276443) at the University of Texas MD Anderson Cancer Center. We used RNA-Seq analysis of MpBC patient tumor biopsies, acquired at the time of enrollment in the clinical trial, to generate a human TRIM24 metaplastic breast cancer signature using the SIGNATURE system, as previously described[41], and developed a heatmap representation of the TRIM24 metaplastic signature expressed by murine metaplastic carcinosarcoma (M) versus carcinoma tumors (C) (Fig. 6a and Supplementary Data 8). Hallmark pathway analysis showed that the TRIM24 MpBC signature exhibit significant up-regulation of Glycolysis, EMT, E2F, and mTORC1 signaling pathways (Fig. 6b). This analysis shows a consistent overlap at gene expression level between human MpBC tumors and TRIM24-driven metaplastic carcinosarcoma.

Global expression profiles of human TNBC tumors, both MpBC and non-metaplastic cases of the ARTEMIS cohort, were assessed by our generated TRIM24 metaplastic signature and found significantly higher in MpBC TNBC but not non-MpBC TNBC (Fig. 6c). Further, TRIM24-signature-high-scoring MpBC tumors (red dots) also showed a higher percentage of vimentin IHC staining, indicative of EMT, compared to non-metaplastic TNBC tumors (Fig. 6d and Supplementary Data 9). Patients with TNBC, including MpBC, are subjected to chemotherapy of anthracyclines followed by taxanes, as standard-of-care primary therapy. Residual cancer burden (RCB) is calculated after neoadjuvant therapy to predict disease recurrence and patient survival, hence, a higher RCB index correlates with worse patient survival[42]. RCB-II and RCB-III tumors had higher TRIM24 scores, but did not achieve statistical significance due to low numbers of chemosensitive tumors (Fig. 6e and Supplementary Data 9), indicating little to no response to primary chemotherapy. In the subtypes of TNBCs, described by Lehman et al.[43], we found that MpBC tumors with significant TRIM24 metaplastic scores were classified mainly as immunomodulatory (IM) and mesenchymal (M*) subtypes (Fig. 6f and Supplementary Data 9). Enrichment of Mesenchymal subtypes is highly consistent with the association of TRIM24-driven tumors and up-regulation of EMT[43]. Overall, these results suggest that TRIM24-driven mouse metaplastic carcinosarcomas exhibit significant similarities to human MpBC tumors by gene expression and hallmark pathway profiling, offering an animal model of human triple-negative MpBC suitable for development of potential therapeutics.

**PROTAC targeting of TRIM24 decreases tumor cell viability.** TNBC, both non-MpBC and MpBC, patient-derived xenograft (PDX) mouse models were established with patient biopsies obtained through the ARTEMIS trial. Non-MpBC and MpBC subtyped patient biopsies were further classified by Vanderbilt-Based subtyping (Supplementary Fig. 5a). PDXs were created by embedding human patient needle biopsies in mammary fat pads of nude mice, which were "humanized" with immortalized human mammary stromal fibroblasts[44]. Implanted and established PDX tumors were grown and passaged two times in nude mice prior to collection of PDX tumors from passage 3 for analyses[44]. Human mitochondrial protein IHC for nonglycosylated components of mitochondria distinguishes human tissue in the mouse mammary environment of PDX models (Supplementary Fig. 5b, d). PDX tumors were further analyzed by TRIM24 IHC to identify positive TRIM24 expression (Fig. 7a). Both MpBC and TNBC (non-MpBC) PDX tumors are positive for TRIM24 expression. MpBC PDXs have high levels of TRIM24 expression in both nuclear and cytoplasmic compartments, while non-MpBC TNBC PDXs have more variable TRIM24 expression (Fig. 7a and Supplementary Fig. 5c).

Previously, our collaborative efforts yielded a high-affinity, small molecule inhibitor of the TRIM24 bromodomain, IACS9571[20]. IACS9571 proved to be highly effective in displacement of TRIM24 from chromatin, as measured biochemically in vitro[20] and by loss of TRIM24 SUMOylation in cellulo[45], but had little effect on proliferation of established breast and other cancer cell lines grown as 2-D cultures. However, IACS9571 bromodomain inhibitor was further developed as a VHL-ligand-based proteolysis targeting chimera (PROTAC), which proved to be an effective TRIM24 protein degrader (dTRIM24) in cellulo ([19] and Supplementary Fig. 5e). In this previously reported study, dTRIM24 added at 5 μM concentration inhibited growth of leukemia-origin, cultured MOLM-13 cells but had little effect on the only breast cancer-origin cells tested, MCF7, a Luminal A breast cancer-derived cell line[19]. The bioavailability of dTRIM24 for in vivo assessment remains undocumented; thus, we assayed dTRIM24 (kind gift of L. Gechijian, N. Gray and J. Bradner) and its chemical enantiomeric negative control (eTRIM24) using TRIM24-driven primary carcinosarcoma cell lines and control cell line 823. The dTRIM24 drug successfully degraded TRIM24 compared to eTRIM24 and DMSO (Supplementary Fig. 5e). To ascertain potential relevance of TRIM24 targeting in human TNBC, we compared the ability of dTRIM24 and eTRIM24 to impact viability of TNBC PDX tumorspheres, both MpBC and non-MpBC (Fig. 7a and Supplementary Fig. 5a), using a high-throughput drug screening platform (Fig. 7b). We found that treatment with dTRIM24 led to significant loss of cell viability for all four MpBC PDX tumorspheres and two

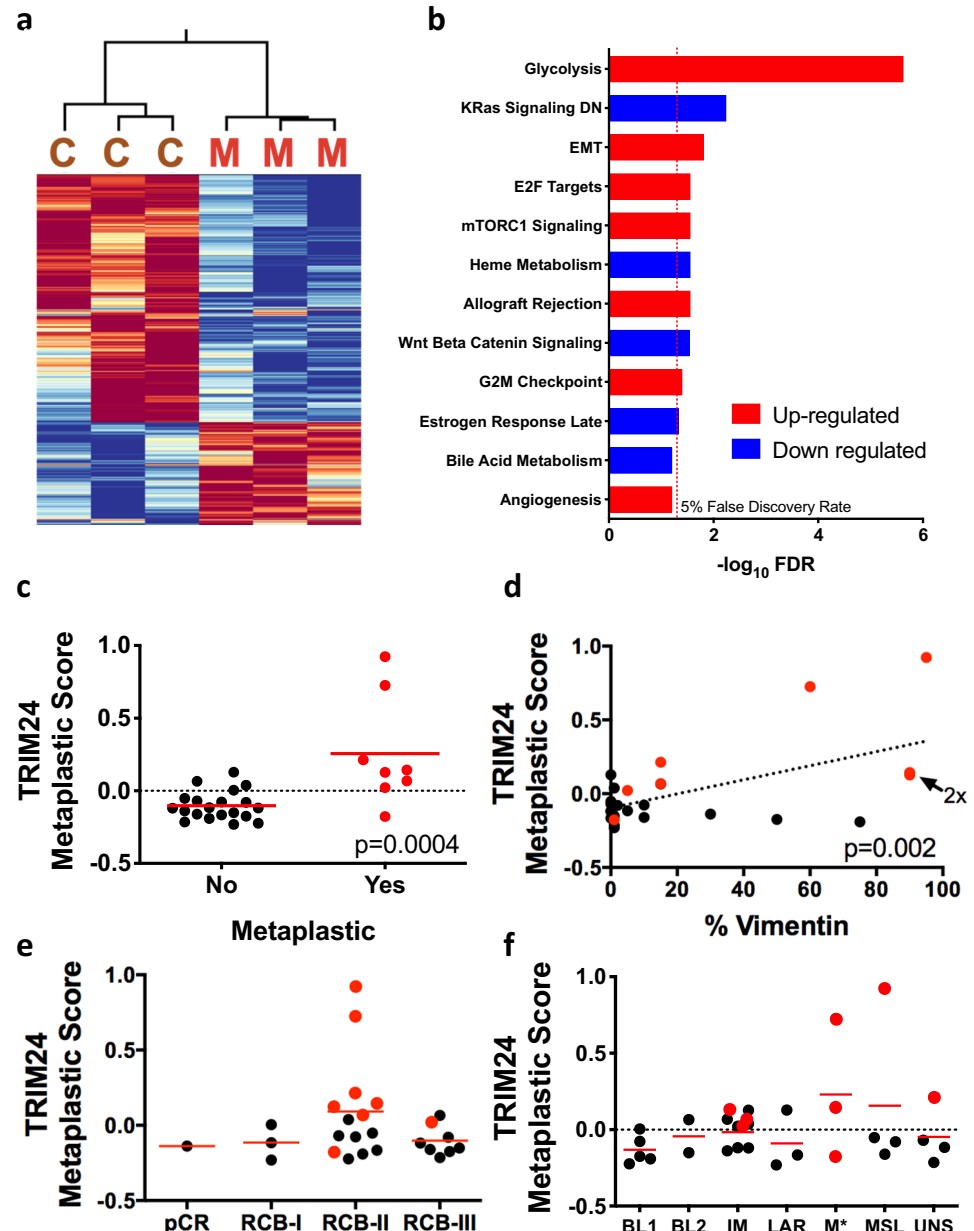

**Fig. 6 A murine TRIM24 metaplastic carcinosarcoma gene signature correlates with human TNBC MpBC patient signature, EMT, and chemorefractory mesenchymal subtype.** **a** Heatmap of expression of genes from the TRIM24 signature in *Trim24^COE* tumors and human metaplastic TNBCs. C - *Trim24^COE* carcinomas and M - *Trim24^COE* metaplastic carcinosarcomas. **b** Statistical significance of Hallmark pathways most strongly enriched in the TRIM24 signature. Red: up in TRIM24, Blue: down in TRIM24. The dotted line represents FDR < 0.05 threshold. **c** TRIM24 metaplastic score matched to MpBC patient gene expression (red) compared to non-MpBC TNBC patient gene expression (black). **d** Correlation of percentage of Vimentin staining in patient biopsies from the ARTEMIS TNBC cohort. MpBC tumors are in red. **e** Association of TRIM24 scores with response to primary chemotherapy in TNBC and MpBC patients from the ARTEMIS trial. pCR: pathological complete response, RCB-I-III – residual cancer burden categories 1–3. MpBC patients are shown in red. **f** Distribution and classification of the ARTEMIS patient cohort and their correlation with TRIM24 metaplastic score to various subtypes of TNBC. BL basal like, IM immunomodulatory, LAR luminal androgen receptor, M* mesenchymal, MSL mesenchymal stem-like, UNS unstable tumor type. Data represented as mean in **c**–**e** and *p*-values are calculated based on two-tailed unpaired *t*-test.

of four non-MpBC TNBC PDX tumorspheres, versus negative control eTRIM24. Overall, these results suggest that targeting TRIM24 or TRIM24-regulated pathways may have therapeutic potential in the future for patients with TNBC, especially the rare, aggressive and chemoresistant MpBC subtype.

## Discussion

Here, we present a transposon-mediated, conditional expression *Trim24* mouse model and show that increased mammary-specific expression of TRIM24 (*Trim24^COE*) drives mammary tumorigenesis. Unexpectedly, a majority (67%) of *Trim24^COE* tumors are carcinosarcomas with coexisting malignant epithelial and spindle (mesenchymal) cellular components, a defining characteristic of human breast carcinosarcoma or MpBCs[11]. Metaplastic breast carcinomas (MpBCs) are a rare, aggressive, chemorefractory and understudied subtype of TNBC. With few effective treatment options for these patients, it is essential that the molecular mechanisms governing the development of these

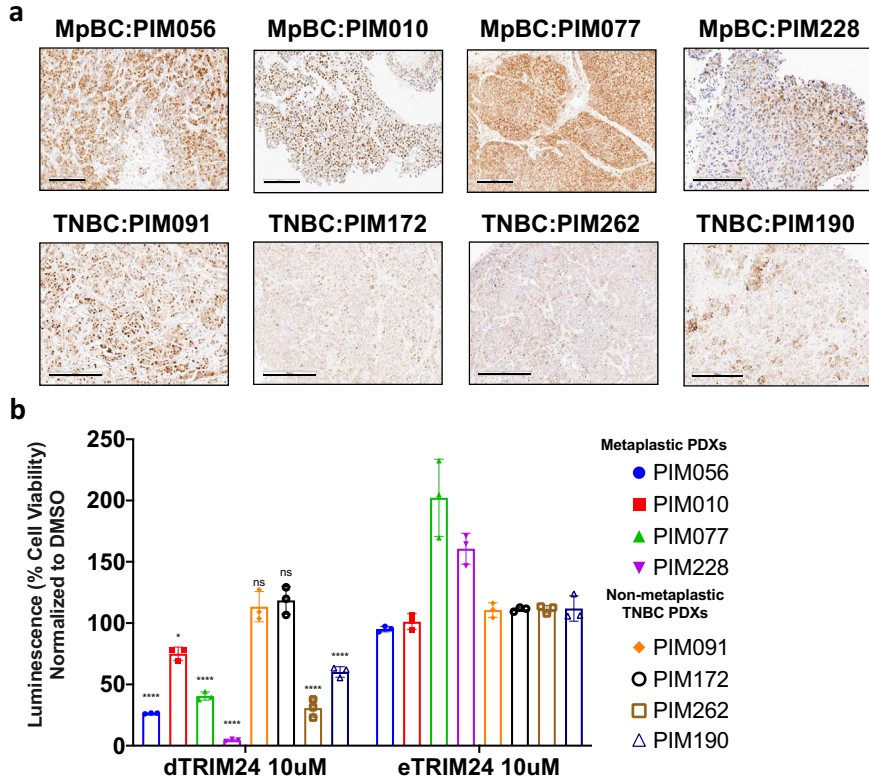

**Fig. 7 Human MpBC patient-derived xenografts (PDX) overexpress TRIM24 and show reduced cell viability when treated with a TRIM24 protein degrader. a** Representative images of TRIM24 IHC of human MpBC PDXs (PIM056, PIM010, PIM077, and PIM228) compared to control samples, non-MpBC-TNBC PDXs, (PIM091, PIM172, and PIM262). Scale bar = 200 μM for PIM056, PIM010, and PIM077. 300 μM for PIM228 and all TNBC PDXs. **b** Representative graphs show luminescence-based assays of cell viability with TRIM24 degrader (dTRIM24) or negative control eTRIM24 treatment of human MpBC PDX-derived suspension cells compared to TNBC non-metaplastic PDX cell suspensions. The percentage cell viability is calculated by normalizing luminescence to DMSO treatment. Significance is calculated using ANOVA between eTRIM24 and dTRIM24 of each sample ($n = 3$ technical replicates). Data represented as mean ± SEM and $p$-values are calculated by two-way ANOVA for multiple comparison using Holm-Sidak method (*$p = 0.0351$, ****$p < 0.0001$, ns non-significant).

rare tumors be understood. In a cohort of 46 MpBC patients, we showed that nuclear TRIM24 is associated with poor overall survival; whereas, cytoplasmic TRIM24 is not. Cytoplasmic TRIM24 reportedly ubiquitinates TRAF3 and facilitates antiviral immunity[4,23], a function that may not be relevant to MpBC tumor development and patient survival.

In the current study, we focused on nuclear roles of TRIM24, as a histone reader and transcriptional regulator in MpBC. Differential gene expression analysis identified glycolysis and EMT as the top up-regulated pathways in TRIM24-driven mammary tumors, robustly active even in end-stage tumors, analyzed here. Our previous *in cellulo* studies showed that immortalized human mammary epithelial cells were transformed and formed high-grade, xenograft tumors when *TRIM24* was expressed by lentiviral transduction[7]. Nanostring analyses of genes differentially expressed in TRIM24-transformed cells showed similar patterns of up-regulated glycolysis and mTOR pathways and cell cycle check point alteration. These data, generated both in cellulo and in vivo, suggest that high levels of TRIM24, in addition to reprogramming epithelial cells to a mesenchymal phenotype, promote an environment in which transformed cells alter metabolic response to survive and proliferate during tumorigenesis.

We developed a TRIM24-driven metaplastic carcinosarcoma gene signature and compared it to human MpBC tumor global gene expression to derive a TRIM24 metaplastic score for each MpBC patient in the cohort reported in this study. Gene expression profiles of human MpBC tumors exhibit significant

EMT expression signatures, consistent with the presence of epithelial/mesenchymal compartments within these highly aggressive tumors[12], and loss of E-cadherin and gain of robust vimentin expression in *Trim24*$^{COE}$ carcinosarcomas, reported here. Glycolysis and EMT are linked in many cancer types[46], and it has been shown that up-regulation of glycolysis modulates the actin cytoskeleton, which may promote EMT[47]. Molecular subtyping of TNBCs shows an enrichment of EMT-associated gene expression in mesenchymal and mesenchymal stem-like subtypes, 10–30% of which are classified as MpBC by light microscopy[12]. TNBC tumors with high TRIM24 metaplastic signature scores are mainly IM and M subtypes and exhibit high vimentin expression. These MpBC tumors also showed poor outcomes in response to systemic chemotherapy (anthracyclines followed by taxanes) in a neoadjuvant setting (NACT), which is reflected by their extensive residual disease (RCB-II and RCB-III) status (Fig. 6e). A recent in-depth proteomic landscape of MpBC patients further defined subtypes of MpBC and compared MpBC to the broader TNBC classification[18]. This cutting-edge report revealed that MpBC patient samples, as well as spindle MpBC-like tumors that develop in a mammary epithelial-specific *Ccn6* knockout mouse[48], have enrichment in EMT and E2F pathways at protein levels consistent with our transcriptomic analysis of both MpBC TRIM24 signature and TRIM24-driven metaplastic carcinosarcoma tumors. This subtype classification additionally underscores heterogeneity of MpBC tumor development and further need for multiple MpBC mouse models, as down-regulation of

CCN6 and cellular metabolism and protein translation up-regulation are not significant in our epigenetic-driven model. Our analyses here are limited to TNBCs. While previous studies and public data sets showed that *TRIM24* RNA is expressed across a broad range of cancers, it is not yet established whether TRIM24 plays a similar role in promoting malignancy through alterations of EMT, E2F, and metabolic pathways. The use of our TRIM24 signature may partially address this issue, although tissue- and batch-specific differences across gene expression data sets may confound the analyses.

Our molecular analyses of *Trim24*COE metaplastic carcinosarcoma tumors nominated c-MET receptor tyrosine kinase, an upstream activator of PI3K/AKT/mTOR pathways, as a potential therapeutic target. A mechanism for increased *Met* expression in *Trim24*COE metaplastic carcinosarcomas lies in specific enrichment of PHD/bromodomain protein TRIM24, a co-regulator of transcription, at accessible chromatin of the activated *Met* promoter, consistent with TRIM24-induced RNA expression and downstream activation of the PI3K and mTOR pathways. Interestingly, a previously reported mouse model with MMTV-driven *Met*mt developed a high percentage of claudin-low, mixed pathology mammary carcinomas that undergo EMT, and a low percentage of adenocarcinoma and adenomyoepithelial carcinoma[30]. Comparison of RNA expression changes in MMTV-driven *Met*mt tumors, as well as to publicly available human MpBC (GSE57544), revealed significant concurrence with our derived TRIM24 Metaplastic signature score (Supplementary Fig. 3d, e). Although we did not see activation of downstream PI3K genes by RNA expression analysis of TRIM24-driven tumors, CyTOF and RPPA analysis showed post-transcriptional activation of PI3K pathway components, consistent with c-MET kinase functions. Previously, Coussy et al.[17] showed that a combination of PI3KCA and MEK inhibitors significantly reduces tumor growth in a limited number of MpBC PDX models. To inhibit the PI3K pathway, we used multiple approaches and found that cell viability of primary *Trim24*COE metaplastic carcinosarcoma cells was reduced in a dose-dependent manner by inhibiting either c-MET, with Critzotinib and PHA-665752, or PI3K/mTOR, with dual inhibitor Dactolisib. Crizotinib is an FDA-approved drug for metastatic NSLC and ALCL; whereas, PHA-665752 and Dactolisib are bioavailable and have been tested in clinical trials for various cancers although not reported as potential therapeutic treatments of MpBC.

Knowing that single-agent therapeutics often lead to resistance and relapse in human cancers, we pursued assessment of TRIM24 inhibition and degradation, using a previously reported PROTAC version of TRIM24 bromodomain inhibitor IACS-9571[19]. PROTAC-TRIM24 [IACS-9571] (dTRIM24) was tested for inhibition of human MpBC PDX tumorsphere viability, as a surrogate for MpBC patient tumor samples. dTRIM24 was previously shown effective in inhibition of a cultured leukemia cell line (MOLM13) viability but, to our knowledge, has not been tested for inhibition of cell viability in solid tumor PDXs[19]. Recent advances in drug discovery have shown the potential of PROTACs for therapeutic success, such as ER PROTAC ARV-471 in clinical trials for ER-positive breast cancer[49]. We propose that combinatorial therapeutics featuring either a c-MET or dual PI3K/mTOR inhibitor in combination with epigenetic targeting, via a bioavailable TRIM24 degrader, may show promise for treatment of MpBC and other TRIM24-expressing TNBC patients with otherwise poor prognosis and survival.

Our studies establish TRIM24 as a potent oncoprotein in mammary epithelia, acting as a histone reader in chromatin association and disruption of metabolic homeostasis, as well as activation of EMT. Similar to human MpBC patients, our TRIM24-driven mouse model also shows activation of MET/PI3K pathways, which can be therapeutically targeted. Importantly, we observed a TRIM24 signature in a rare and aggressive form of human breast cancer, MpBC. Preclinical studies are needed to validate that inhibiting TRIM24 and its downstream targets, such as c-MET and PI3K pathway activities, has therapeutic potential for this specific subset of TNBC patients.

## Methods

**Mouse model**. A *piggyBac* transposon vector was constructed bearing a human *UBC* promoter-driven loxP-lacZ-stop-loxP cassette 5′ of a FLAG-tagged *mTrim24* cDNA (pBac-UbC-lox-stop-Trim24) (Supplementary Fig. S1a and Supplementary Table 1)[50]. Uncut *pBac-UBC-lox-stop-Trim24* (2 ng/μl) was injected into the pro-nuclei of FVB/NCr (Charles River Laboratories) zygotes along with transposase RNA (15 ng/μl). The FVB/NCr strain is reportedly more permissive for mammary tumor development[51]. Transgenic *Trim24* (*Trim24*LSL) founder mice were identified by PCR of tail genomic DNA (Fig S1b), using primers listed in Supplementary Data 1. *Trim24*LSL mice were bred with the Tg(MMTV-cre)4Mam (The Jackson laboratory, stock 003553) recombinase mouse line D (*MMTV-Cre*Tg/0)[52] for Cre-mediated excision of the floxed stop cassette to drive expression of transgenic *mTrim24* in mammary epithelial cells. The resulting mice were called *Trim24*COE and maintained in pathogen-fee conditions. We also maintained an aging virgin female colony by breeding MMTV-Cre mice with FVB to generate control mice of similar background as *Trim24*COE and *Trim24*LSL mice. Virgin female mice were aged until there was evidence of a palpable tumor or maximum of 2 years. At killing, mammary glands, liver, lung, and spleens were dissected and preserved for further analyses. Tumor penetrance was calculated as percentage of mice in tumor bearing genotype *Trim24*COE divided by total number of mice in *Trim24*COE cohort. Animal handling and experiments were performed as approved by the Institutional Animal Care and Use Committee at the MD Anderson Cancer Center.

**Whole-genome sequencing**. To determine the genomic location of transgenes, we performed whole-genome sequencing using tail tissue-derived DNA of three different *Trim24*LSL progeny of the same founder. The library preparation and sequencing were done at the MD Anderson Next Generation Sequencing Core Facility. Briefly, the libraries were prepared using a KAPA Hyperprep kit, Roche (KK850) and samples were run for 76 bp paired-end to achieve 5–10x coverage per sample. The sequence reads were mapped by Burrows-Wheeler Aligner (BWA) (version 0.7.15)[53] to the mouse genome (version GRCm38). In order to detect possible locations of *Trim24* transgene, we included the *Trim24*LSL vector sequence into the mouse genome as an extra "chromosome". The mapped bam files were generated by GATK pipeline 3.7 developed by the Broad Institute[54]. The discordantly mapped reads with Trim24 gene sequences were then retrieved by samtools[55]. The discordant reads of these samples are included in Supplementary Data 3.

**Carmine alum staining**. Inguinal mammary glands were dissected from littermate female mice and compared only between mice of the same age and harvested at the same time of day. Mammary glands were flattened onto a glass slide and stained as previously described[52] with the following alterations: mammary glands were fixed in ice cold 4% paraformaldehyde in phosphate-buffered saline (PBS) overnight, washed in PBS 3 × 10 min then submerged in carmine alum staining solution (4 mM carmine, 10 mM aluminum potassium sulfate) until staining thoroughly permeated the tissue. Tissues were then dehydrated as described[52], cleared in Histo-clear (National Diagnostics) overnight and stored in toluene. Cleared mammary glands were imaged using a stereomicroscope. To count the number of branches, one major branch was selected, and individual sub-branches were counted manually. The graph was generated using Prism (v9) and student t-test was performed to calculate significance between control and *Trim24*COE mammary branches.

**Immunohistochemistry**. All hematoxylin and eosin (H&E) staining was performed using standard procedures by the MD Anderson Department of Veterinary Medicine Histology Core. IHC for Flag (Sigma,sc-7945, 1:5000), ER (Santa Cruz, sc-8002,1:500), PR (Abcam, ab63605,1:4000), E-cadherin (Proteintech, 20874-1-AP, 1:100) Vimentin (Abcam, ab92547, 1:500), human mitochondria (Chemicon # MAB1273, 1:2000), and TRIM24 (Proteintech,14208-1-AP, 1:500) were performed by the MD Anderson Cancer Center Research Histology Pathology and Imaging Core using protocols as previously described[7]. Forty-six archived MpBC paraffin tissue blocks from the Department of Pathology at MD Anderson Cancer Center, with up to 22 years of clinical follow-up data, were prepared for TRIM24 IHC using the standardized departmental protocols. The tissues were scored as previously described[22]. Next, patients were stratified based on scoring and percentage of total cells expressing TIRM24 either in nucleus or cytoplasm or both. The group of patients were divided into four groups and overall survival was calculated. Statistical analyses were carried out using GraphPad Prism (version 8) software. The cumulative overall survival was calculated using the Kaplan–Meier method, and log-rank test was used to analyze differences in the survival times.

**Protein and RNA analyses.** To prepare whole cell lysates, cells were lysed as indicated in either radioimmunoprecipitation assay (RIPA) buffer (20 mM Tris pH 8.0, 150 mM NaCl, 1 mM EDTA, 1% NP-40, 1, Triton X-100, 0.5% deoxycholic acid or NETN buffer (150 mM NaCl, 1 mM EDTA, 50 mM Tris pH 7.5, 0.1% NP-40) or 1% SDS supplemented with protease inhibitor cocktail (EMD Millipore). Immunoblotting was performed using standard techniques for antibodies listed in Supplementary Data 1. Total RNA was isolated with TRIzol reagent (Invitrogen) and qRT-PCR was performed as described previously[7] using primers listed in Supplementary Data 2. Data were calculated using $2^{-\Delta\Delta CT}$ method in Microsoft Excel (365) and graphs were prepared using Prism (v8).

**Primary cell lines.** Tumor fragments designated for cell isolation were harvested as previously described[56]. Briefly, tissue sections were minced and placed in 10 ml digestion media (Dulbecco's Modified Eagle Medium (DMEM)-F12, 250 u/ml hyaluronidase, 3 mg/ml collagenase, 100 U/mL penicillin/streptomycin (PS), 100 μ/mL antibiotic-antimycotic) per gram of tissue. Tissues were digested while shaking at $1500 \times g$ for 1–3 h at 37 °C, filtered through a 40 μm sterile filter and centrifuged at $1200 \times g$ for 10 min at 4 °C. Cells were resuspended in DMEM/F12 and washed 4X sequentially by centrifugation at $1200 \times g$ for 5 min. The resulting suspensions were plated onto 60 mm tissue culture dishes in complete media (DMEM 10% fetal bovine serum (FBS), and PS) at 37 °C. Cells were grown to confluence, during which time observable colonies were formed. These cell colonies were then subjected to differential trypsinization, removed and gently tapped to release colonies enriched for epithelial cells, leaving behind mesenchymal and fibroblast-like cells. The resulting population was termed epithelial-enriched and then plated onto a new dish and cultured in complete media. Three primary cell lines, 567, 64, and 897, were generated from TRIM24-driven metaplastic carcinosarcoma and cultured using complete media. Primary cell line 823 was generated from a spontaneous mammary tumor generated in an *MMTV-Cre* mouse and used as a control. Cells were passaged and frozen after Mycoplasma testing (Lonza MycoAlert Mycoplasma Detection kit, LT07218) as per manufacturer's instruction. The passages of primary cells used in all experiments were between 8 and 15. For tet-inducible shRNA cell lines, we used a lentivirus system and Lipofectamine 3000 (ThermoFisher Scientific, L3000008) to transfect 293FT cells using manufacturer's recommendations on virus production. Both shControl (Horizon, Catalog No. RHS4743) and shTrim24 (Horizon, Catalog No. RMM4431-200368247,) were purchased through the functional genomic core at University of Texas M.D. Anderson Cancer Center for cloning in a TRIPZ inducible lentivirus vector. Next, lentivirus was collected from 293FT cells to transduce the TRIM24-driven primary metaplastic carcinoma cell line 897. The shControl and shTRIM24 897 cells were grown under puromycin selection and sorted based on RFP after tetracycline induction (1 μg/ml). To collect RNA, 897 sh-lines were induced with tetracycline at 1 μg/ml for 24 h. Cells were tested for pathogens using IMPACT testing (IDEXX BioResearch) prior to inject into mouse for generating allograft tumors. Mouse mammary tumor-derived 4T1 cell line was purchased from ATCC (CRL-2539) and grown in RPMI-1640 media.

**RNA library preparation and deep-sequencing.** To perform deep sequencing of RNA (RNA-seq), total RNA was isolated from individual tumors, in parallel with freshly isolated, disease free mammary glands (controls), using Zymo Direct-zol RNA miniprep kit (R2050). RNA libraries were prepared using TrueSeq RNA Library Prep Kit v1 (Illumina) followed by sequencing at the MD Anderson Next Generation Sequencing Core Facility.

**RNA-seq analysis.** Pre-preprocessing and analysis of RNA-Seq data used pipelines developed by BETSY[57]. To summarize, we checked the quality of the FASTQ files using FastQC (v 0.11.8). Then, we trimmed adapters with Trimmomatic (0.39) using default settings[58]. Trimmed reads were checked again with FastQC, and then aligned to the GRCm38 mouse genome assembly with the GENCODE Release M14 transcriptome using the STAR aligner for *MMTV-Cre*[Tg/0] mammary gland[59]. The reads were counted with HTSeq-Count[60] using the union mode to resolve multiple features. Gene expression values were estimated as TPM using RSEM (version 1.3.1), using the STAR (2.7.2b) aligner option[61]. We evaluated the quality of the alignments using the CollectAlignmentSummaryMetrics and CollectRnaSeqMetrics tools from Picard (2.9)[62]. The data are submitted on Gene Expression Omnibus (GEO) with accession GSE179036 for murine samples.

To determine gene expression profiles of TRIM24-driven tumors, a differential expression analysis was performed to compare profiles of tumors with non-tumor mammary tissue using DESeq2 (3.13)[24]. We used script developed in house to calculate the average expression of tumors and control groups, the differences and fold change expression between groups. Genes were selected using a false discovery rate cutoff of 5% and at least a 5-fold change between groups. This resulted in a list of 2006 genes (Supplementary Data 6). Hallmark pathways[63] were scored on gene expression profiles (read counts) using the ssGSEA algorithm (2.0). Differential pathways were identified using a Student's t-test correcting for multiple hypotheses and accepting pathways with a 5% FDR cutoff.

The association between *Trim24*[COE] tumors and a previously established metaplastic gene expression signature were calculated using GSEA (4.1.0)[25]. The

LIEN_BREAST_CARCINOMA_METAPLASTIC_VS_DUCTAL_UP gene set was downloaded from the MSigDB database.

**Reverse Phase Protein Arrays analysis.** Flash-frozen tumor samples and mammary glands were submitted to the Functional Proteomics RPPA Core Facility at MD Anderson Cancer Center. The samples were processed as described previously[31]. Samples were probed with 244 antibodies listed in Supplementary Data 7. The staining intensity was normalized to an internal loading control and customized software to produce Normalized Linear values and transformed Log2 values. The normalized raw values were used to generate a heatmap using R function. The protein names were transformed to gene names to calculate GSEA scores, subjected to Hallmark pathway analysis and plotted using Prism (v8)[25,63].

**Cytometry by time of flight.** To validate RPPA analysis, we generated allograft tumors by injecting primary cell lines derived from TRIM24 overexpressing tumors into mammary fat pads of *Foxn1*[nu] mice, also known as athymic mice[32]. Tumors were dissected once they reached 2 cm³ in size. We also collected normal mammary glands of MMTV-Cre[Tg/0] mice as controls. The samples were prepared and processed for CyTOF as previously described[64]. FlowJo_v10.3 was used to remove beads, debris and to obtain single cells. CytoBank (version 6.7) was used to generate viSNE plots.

**c-Met and PI3K/mTOR inhibitor treatment.** MET inhibitors Crizotinib and PHA-665752 were purchased from Cayman Chemical (Item numbers 12087 and 14703, respectively). A dual inhibitor of PI3K and mTOR, Dactolisib or NVP-BEZ235, was also purchased from Cayman Chemical (Item number 21185). Primary murine cells obtained from TRIM24-driven carcinosarcomas (567, 64, and 897) and a TNBC cell line (4T1) were plated at a confluency of 2000 cells per well in 96-well plates. Each drug of choice or vehicle control (Dimethyl sulfoxide or DMSO) were added the next day and incubated for 72 h. At the end of 72 h, post-treatment cell viability assays were performed.

**Cell viability.** Cell viability was measured using CellTiter-Glo® (CTG) Luminescent Cell Viability Assay (Promega G7570), as per manufacturer's recommendations. On the last day of treatment, CTG reagent was added at a 1:1 ratio, as per manufacturer's recommendation. Viability was measured using luminescence on a FLUOstar Omega platform as per manufacturer's guidance.

**TRIM24 metaplastic signature.** We scored the TRIM24 metaplastic signature on a panel of 68 triple-negative human breast cancer tumors from the ARTEMIS trial (NCT02276443) that have been profiled by RNA-Seq. They were processed using a similar pipeline as above, except that they were aligned to the human reference genome hg19 and a transcriptome from RefSeq[65]. To compare the mouse-derived metaplastic signature with the gene expression profiles of human tumors, we converted the signature to human genes based on the homologs in the Homologene database. We then scored the signature on human tumors using the SIGNATURE system, as described previously[41]. We then compared the scores between the metaplastic tumors (defined by a pathologist) and the non-metaplastic ones using a two-tailed T-test via GraphPad Prism (Supplementary Data 8). We also compared the scores against the vimentin percentage as determined by IHC on tissue slides and calculated the *p*-value using a Pearson test as implemented in the R programming language. The data are submitted to GEO with accession number GSE165407.

**TRIM24 degrader treatment.** Human MpBC PDX tumors were harvested and cells were extracted in cell suspension media as previously described[44]. These cells were distributed on 384-well plates and subjected to a high-throughput robotic imaging and drug delivery platform (Texas A&M Combinatorial Drug Discovery Program, Houston, TX). The drugs used for treatment were eTRIM24, a negative control, and dTRIM24, a degrader of TRIM24, and diluted to 10 mM in DMSO[19]. DMSO was used as a vehicle control. Cell viability was measured using CellTiter-Glo. The measured cell viability values were normalized to DMSO controls and plotted as percent inhibition using GraphPad Prism. To determine the protein expression level under the treatment of DMSO, eTRIM24, and dTRIM24, cell lines 823, 567, 64, and 897 were treated with 4μM of eTRIM24 and dTRIM24 along with equivalent amount of DMSO in 6 well plate in triplicates. Treated cells were collected after 48 h for western blot analysis as described earlier.

**ATAC-seq library generation.** ATAC-seq in replicates was performed following the Omni-ATAC protocol as previously described[66]. Briefly, cells grown in tissue cultures were pretreated with 200 U/ml DNase (Worthington) for 30 min at 37 °C to remove free-floating DNA and to digest DNA from dead cells. The cell cultures were washed with PBS followed by harvesting and counting. 50,000 cells were lysed and nuclei isolated after centrifugation at $500 \times g$ for 10 min in lysis buffer containing 0.1% NP-40 and 0.01% Digitonin. Pelleted nuclei were tagmented with Nextera Tn5 Transposase (TDE1, Illumina 15027865) in TD Tagment DNA buffer (Illumina 15027866) for 30 min at 37 °C, and the resulting library fragments were purified using a Qiagen MinElute kit. Libraries were amplified by 4–6 PCR cycles as

described[66] and purified using AMPure XP beads (Beckman Coulter A63881). ATAC-seq libraries were sequenced $2 \times 75$ bp on an Illumina HiSeq4000 to obtain at least 50 million high quality mapping reads per sample.

**ATAC-seq analysis**. The libraries for ATAC-Seq were sequenced using $2 \times 75$ bases paired-end protocol on Illumina HiSeq 3000 instrument. 58–67 million pairs of reads were generated per sample. Each pair of reads represents a DNA fragment from the library. Adapter sequences were removed from 3'ends of reads by Trimmomatic (0.39) using default settings[58]. Then the reads were mapped to the mouse genome (mm10) using Bowtie (version 1.1.2)[67] with parameters "-allow-contain -maxins 2000 -v 2 -m 1 -best -strata". About 89–91% pairs of reads were mapped to the mouse genome, with 84–86% uniquely mapped. To avoid PCR bias for multiple pairs of reads that were mapped to the same genomic position, only one copy was retained for further analysis. After the removal of reads by chrM, 40–46 million pairs of reads were used in the final analysis. For each fragment (e.g. a pair of reads), the 5' end was offset by $+4$ bp and the 3' end was offset by $-5$ bp to adjust both ends to represent the center of a transposon binding event. To generate the signal landscape, binding events (i.e. ends of fragments) were smoothed by 73 bp (i.e. extended 36 bp upstream and 36 bp downstream), piled up and normalized to 10 M total binding events. The resulting values were averaged over every 10 bp window and displayed in Integrative Genomics Viewer (IGV) (2.8)[68]. The data are submitted on GEO (GSE149685).

**Chromatin immunoprecipitation**. ChIP assays were performed, as previously described[5], using murine *MMTV-Cre* expressing tumor primary cell line 823 and TRIM24 overexpressing metaplastic carcinosarcoma primary cell line 897. Additionally, after sonication by a BioRuptor 300, chromatin was further treated with MNase (Worthington, LS004798), 30 min at room temperature. TRIM24 (Proteintech, 14208-1-AP; Bethyl, A300-815A) antibodies were used for immunoprecipitation, and the precipitate was collected by Dynabeads Protein A (Invitrogen, 10002D).

**Reporting summary**. Further information on research design is available in the Nature Research Reporting Summary linked to this article.

## Data availability
The RNA-Seq data generated for TRIM24-driven mouse model (Fig. 3) are deposited in the GEO database under accession code GSE179036. The RNA-Seq data associated with human metaplastic breast cancer patients (Fig. 6) are deposited in the GEO database under accession code GSE165407. The ATAC-Seq data of TRIM24-driven metaplastic carcinosarcoma cell line and control cell line are available at GEO database under accession code GSE149685. WGS data associated with identification of Flag-tagged mTrim24 transposon in transgenic mice are available at figshare repository (Sample 1 R1 file at https://doi.org/10.6084/m9.figshare.14818542.v1(2021), Sample 1 R2 file at https://doi.org/10.6084/m9.figshare.14818560.v1(2021), Sample 2 R1 file at https://doi.org/10.6084/m9.figshare.14818536.v1(2021) and Sample 2 R2 files at https://doi.org/10.6084/m9.figshare.14818575.v1(2021)). MsigDB LIEN_BREAST_CARCINOMA_METAPLASTIC_VS_DUCTAL_UP was used to compare gene expression of TRIM24-driven metaplastic tumors. Source data are available as a Source data file. The remaining data and information associated with this manuscript are available within the article as a supplementary information. Source data are provided with this paper.

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

## Acknowledgements

We thank L. R. Patel and the Lozano lab for their help and valuable input, as well as L. Gechijian, N. Gray and J. Bradner for their kind gift of a TRIM24 PROTAC. This study was supported by National Institutes of Health grant RO1 CA214871 to MCB and GL, the generous philanthropic contributions to The University of Texas MD Anderson Cancer Center Moon Shots Program™, and the MD Anderson Cancer Center National Cancer Institute Cancer Center Support Grant (CCSG) CA016672. We are grateful to the patients who provided tumor biopsies for PDX model establishment. PDX models and derivatives were obtained from the Cazalot Breast Cancer Model Resource, established through a gift from the Cazalot family and from funds from the MDACC Breast Cancer Moon Shot Program. Additional funding sources that supported this work include the Cancer Prevention and Research Institute of Texas (CPRIT) grants RP160710 (to H.P-W., J.T.C. and P.J.D.) and RP150578 (to P.J.D). Transgenic animals were created at the MD Anderson Cancer Center Genetically Engineered Mouse Facility with the support of CCSG (P30CA016672) and R50 (R50CA211121) grants. RNA-Sequencing was performed at MD Anderson Cancer Center Science Park Next-Generation Sequencing Facility supported by CPRIT Core Facility Support Grants (RP120348 and RP170002). IHC was performed with the support of CCSG-funded Science Park Research Histology, Pathology and Imaging Core (RHPI) at University of Texas MD Anderson Cancer Center Science Park, the Flow Cytometry and Cell Imaging Core shared resource and Histopathology core of the Department of Veterinary Medicine & Surgery are partially funded by NCI Cancer Center Support Grant P30CA16672. High-throughput drug screening was performed at the Texas A&M Combinatorial Drug Discovery Program (CPRIT CFSA RP150578).

## Author contributions

M.C.B., J.T.C. and S.L.M. conceived and supervised the study. V.V.S., A.D.D. and S.J. designed and performed experiments. V.V.S. and J.T.C. performed N.G.S. analysis. K.L.A. performed CyTOF experiments. S.A.S., V.V.S., A.D.D, S.J and P.M.K. managed mice, performed sample preparation, data analysis and experiments. C.Y., J.T.C., L.H., M.G. and S.L.M. contributed to patient samples and clinical data. L.H. and M.G. performed pathology and scoring of patient samples. A.J. and Y.L. contributed to ATAC-Seq and data analysis. B.L. performed whole-genome sequencing and analysis. J.J.S. performed N.G.S. S.C., Y.T., X.Z., X.Z., Y.J. and H.P.W. contributed to preparation of PDX samples. C.L.C. and Z.K. synthesized TRIM24 PROTAC. R.T.P., L.G., C.S. and P.J.D. contributed to high-throughput drug screening. M.S. and M.G. performed pathology and scoring of mouse samples. J.P.T., G.L. and R.R.B. contributed to design of the mouse model. All authors contributed to editing and writing of the manuscript.

## Competing interests

S. Moulder has partial support for clinical trials from Pfizer, Genetech, Seattle Genetics, Lilly, Merck, EMDSerono, Bayer, AstraZeneca and Novartis, and no other conflict of interest. All other authors have affirmed no conflict of interest.
