## [Peer Review File · Nature Communications]

REVIEWER COMMENTS

Reviewer #1 (Remarks to the Author):

In this manuscript the authors have overexpressed TRIM24 specifically in the mammary luminal epithelial cells through the use of MMTV-Cre. This resulted in tumors after a long latency with several histological subtypes. The subtypes somewhat resembled human MpBC and they illustrated survival differences in these tumors based on nuclear TRIM24 levels. Comparison of normal mammary to TRIM24 tumors identified a number of genetic pathways, several related to EMT and PI3K. Treating human PDX samples to degrade TRIM24 resulted in a reduction of viability and the authors suggest that there may be therapeutic benefits for TNBC patients.

Overall the manuscript is well written and readily understood with important findings for the role of TRIM24 in breast cancer.

There are major issues that may be relatively easily addressed with additional experiments. First, the experimental design starting with results shown in Figures 3 and 4 is highly flawed. Comparison of normal mammary gland to tumor yielded MANY genetic hits, but the problem is that they are comparing two states (normal and tumor) that are completely different. This was compounded by doing bulk sequencing, not single cell, so they are comparing a mammary gland composed of adipose, stroma and a minimal amount of mammary epithelium to a epithelial dense tumor. Two better options would have been to;

- 1) Include TRIM24 transgenic mammary glands, both early (12 weeks) and from contralateral glands when the primary tumor was excised. This would allow for examination of genes that may be leading to transformation, especially if scRNAseq was used.
- 2) Compare the TRIM24 tumors to other characterized tumors. Microarray, RNAseq and scRNAseq data are all available for other mouse mammary tumor models. Indeed, the comparison to the MET tumors from Dr. Park's group begs the comparison to those tumors.

Setting aside the flaw in the experimental design that can be addressed with additional work, the manuscript was informative. Additional major concerns are as follows:

- 1) 8 week glands are shown in Figure 1 and it appears that branching is increased. Were the mice staged for estrus cycle? This should be done routinely.
- 2) Branching should be quantitated and compared with a statistical test at 8 weeks. This analysis should also be repeated for mice that have developed tumors – as well as for age matched controls.

3) Given that only 40% of mice developed tumors, the KM plot showing controls with only 4 tumors is rather weak statistically. In addition, having only 12 mice develop tumors is exceedingly low, especially when the authors attempt to stratify histological subtypes. Additional tumors should be generated and characterized.

4) The point of Figure 1 Panel C is unclear – why is normal control mammary being compared to a tumor? Especially when the tumor histology is shown again in Figure 2.

5) A trained mouse pathologist should examine the tumors – not specialists in human breast cancer pathology. There are a number of talented mouse mammary pathologists that should be consulted and who would likely change the interpretation. While not a pathologist, the dark arrow in Figure 2 Panel C image i DOES NOT denote stroma. Instead, this appears to be a tumor that is in the midst of being taken over by a spindleoid morphology, reminiscent of EMT. For suggestions of pathologists see the following:

a. Robert Cardiff UC Davis (may be retired / retiring?)

b. Tan Ince Weill Cornell

c. Alexander Borowsky UC Davis

Reviewer #2 (Remarks to the Author):

Shah et al have created a transgenic mouse that conditionally overexpresses TRIM24 in mammary gland epithelial cells. TRIM24 is a chromatin reader protein that modulates p53 and metabolic pathways and that appears to promote cancer. Its expression correlates with poor survival of breast cancer patients. In this manuscript, the authors showed that this new line of mice is predisposed to breast cancer. The majority of tumors were carcinosarcomas, which based on the authors' characterization, serve as a model for multiplastic, triple negative breast cancer in humans. The authors then molecularly characterize both the mouse tumors and human patient tumors and identify c-MET and the PI3K/mTOR pathway as relevant targets that may be used to formulate therapeutic approaches. There are not any great mouse models for any kind of triple negative breast cancer (TNBC), the findings are mechanistic and advance the field, and the findings are of enough general interest to the cancer community to justify publication in a journal such as Nature Communications. That being said, the authors need to concentrate on strengthening the existing data and addressing a number of presentation issues.

Issues with data/conclusions, in order of appearance:

1. The authors do not provide enough detail about the founder mice. Was there one line of founder mice or multiple? If multiple lines, were used, the authors should indicate where each was used. Was there any attempt to map the integration site(s)? If there is only one founder line, this may be an issue if the integration interrupted a critical gene or regulatory sequence.

2. MMTV-Cre can allow expression in tissues other than mammary epithelium. Were tumors observed anywhere other than the mammary gland? If so, it should be reported in a supplemental figure/table. Such results would not detract from what was learned about mammary/breast tumors, but should be reported.

3. There appears to be a discrepancy - tumor penetrance was reported to be only 40%, but the decrease in survival indicates 75% of the mice died early compared to control mice. Does that mean there is a tumor-independent effect on survival? The authors should address this.

Were all TRIM24-COE mice terminated after 400 days while the controls were left for 600 days?

4. Figs. 2E and S2B – The reviewer does not understand why the survival curve for patients with both nuclear and cytoplasmic staining was not shown and cannot be done without duplication, as the authors state. Isn't this a separate cohort of patients?

5. One of the more significant issues is the authors' decision to combine transcriptional analysis of the carcinoma and the carcinosarcoma tumors (Fig. 3). Why not show both separately? The case has been made that these mouse carcinosarcoma tumors are similar to multiplastic patient tumors (MpBC), so why confuse the analysis by doing gene expression analysis on a mix of carcinoma and carcinosarcoma mouse tumors? The heterogeneity of the individual carcinoma tumors further argues against combining analysis of the different tumor types.

6. Another significant issue is the weak evidence for direct activation of c-MET by TRIM24. As presented, there is an increase in c-Met in the TRIM24 overexpressing mouse tumors and an unconvincing argument for direct regulation based on the ATAC-seq and CHIP data (addressed in more detail below). Perhaps the authors can knock down TRIM24 in the overexpressing cells or perform some other experiment to better link TRIM24 manipulation with c-MET expression levels.

The CHIP in Figure 5C is simply not convincing. First, the authors state that mononucleosomes were isolated. Yet the two primer sets are identified as "+307" and "+174", which are separated by less than one mononucleosome-length of DNA. So how could one reasonably expect different results for

the two primer sets unless it were known that there were highly and invariantly positioned nucleosomes at that locus that caused the examined sequences to be incorporated into different nucleosomes? In addition, both primer sets are in regions of enhanced chromatin accessibility, so again, why would one necessarily believe that CHIP results would be different?

7. In Figure 6, the authors suddenly jump to PDX tumors as an experimental system and a PROTAC degrader as the tool for manipulation of TRIM24. This is problematic because of the lack of controls and the modest nature of the data.

(A) It's great that others have reported that the PROTAC degrades TRIM24 in a different cell type, but it really should be shown that TRIM24 levels were reduced/eliminated in the cells that were used for the experiments presented here. Westerns may not be possible, but presumably some form of IHC or ICC is possible.

(B) Even assuming the controls are appropriate, the data presented are weak. 4/4 MyBC PDXs showed an effect, but one was marginal. 2 of 3 TNBC PDX samples showed no effect, but one showed a significant effect. Perhaps the authors can buttress the PDX data with inhibitors or by including more samples or by some other means.

8. There is no indication that any of the -omics data have been placed in an appropriate repository.

Issues with presentation, in order of appearance:

9. The last sentence of the abstract is awkwardly worded. Please clarify.

10. None of the Supplemental tables are easily identified. Each table lacks a label indicating the table number and a title indicating what content is being shown. Please address.

11. lines 131-132 "...injected into the pronuclei of FVB/NCr (Charles River Laboratories) zygotes ..." and lines 138-139, "...by breeding MMTV-Cre mice with FVB...". The authors do not describe the strain nor do they explain why it was chosen.

12. lines 139-140 "Virgin female mice were aged until there was evidence of a palpable tumor." Where were tumors located? Were they all mammary tumors? If not, what was observed and which tumors were used for subsequent analyses?

13. lines 194-199 need to be corrected for grammar.

14. Fig. 1D and Supp Fig. 1E should be combined and presented together. It's an IHC staining – why does the reader have to go to the supplement to see the control so that the conclusion can be evaluated?

15. The negative results in Fig. S2A require images of the positive control run with these tumors. Similarly, Fig. S5A requires a negative staining control.

16. Fig. 3B/D lack a scale relating the colors to quantification.

17. There are labeling issues in Figs. 3 and 6. Fig. 3 shows a transition from referring to the mouse carcinosarcoma tumors to referring to them as metaplastic tumors, with the abbreviation of “M”. While by definition this is technically correct, it causes confusion with data from metaplastic breast cancer patients. In Figs. 4 and 5 there is a resumption of use of “carcinosarcoma” for labeling. To be consistent and to minimize confusion, the reviewer recommends that “carcinosarcoma” be used in Fig. 3.

In Fig. 6, the authors again refer to the mouse carcinosarcoma tumors as Metaplastic (M). But then in Fig. 6F, “M” stands for “mesenchymal”. Again, it might decrease confusion if “carcinosarcoma” were used to refer to the mouse mammary tumors, “M” were used for the human tumors, and a different abbreviation were used for “mesenchymal”.

18. line 516-517 reads “..., which was significant only for MpBC (Fig. 6C).” This is open-ended, and should be reported as “significant for MpBC but not non-myeloplatic TNBC”, or similar.

19. line 521 introduces “RCB-II and RCB-III outcomes”. “RCB” is not defined, the “outcomes of II and III” are not defined, and the significance of these outcomes is not defined.

20. Would the graph in Fig. 6A be better presented with TNBC and MpBC as labels rather than No and Yes? Or perhaps consider putting a key within Figs. 6C-F to explain that the red and black dots are MyBC and TNBC patients.

21. line 549. What are MOLM-13 cells and why are they relevant?

Reviewer #3 (Remarks to the Author):

The manuscript is based on preclinical data showing that transgenic mice over-expressing Trim24 develop mammary tumours that resemble to metaplastic carcinomas. Global and single-cell tumor profiling of these murine tumors revealed Met as a direct oncogenic target of TRIM24, leading to aberrant PI3K/mTOR activation. Pharmacological inhibition of these pathways in primary Trim24 tumor cells and TRIM24-PROTAC treatment of MpBC PDX tumorspheres revealed the therapeutic potential of targeting TRIM24.

The findings are new and present some clear interest from a molecular and therapeutic point of view for metaplastic breast cancers.

The major limitations are that the human relevance is not sufficiently explored and in vivo preclinical experiments are lacking to validate the therapeutic potential of TRIM24 targeting.

1. Figure 2A shows the frequency of the different types of sarcoma developed in transgenic mice. How many mice and how many tumours have been analysed in total?

2. The clinical and histological characteristics of the 46 human metaplastic breast cancers used to validate TRIM24 expression are not fully described. Which types of metaplastic breast cancers are analysed (chondroid, spindle, squamous?) Moreover only 10% seem to over-express TIM24. This is an important point to validate Trim24 overexpression in MBC.

3. Additional cohorts of human TNBC and MBC should be analysed to provide evidence of TRIM24 over-expression is a characteristic of metaplastic breast cancer. In silico analysis of public datasets such as cBioportal should also be analysed (the TCGA dataset in cBioportal includes a group of metaplastic breast cancer).

4. What drives TRIM24 over-expression in human breast cancer? An analysis in cBioportal could be performed to analyse TRIM24 expression and gene amplification/gains in different breast cancer

datasets. If TRIM24 over-expression is driven by gene amplification, is there an enrichment of TRIM24 amplifications in metaplastic breast cancers?

5. The analysis of TRIM24 expression in PDX models is done without a statistical analysis: how many TNBC PDX were analysed? Only 1 is shown in figure 7A. Ideally different groups of TNBC should be analysed (basal-like, LAR, mesenchymal tumors , etc.).

6. The data presented in figure 7B are not sufficient to conclude that TRIM24 is a therapeutic target. What is the expression of TRIM24 in the 3 TNBC PDX? are they all negative? In vivo studies should be performed in different PDX models of metaplastic breast cancer.

7. Figure 6F does not show a clear enrichment of the TRIM24 signature in the MSL subtype. Moreover, in the M subtype there are many tumors that are not metaplastic breast cancers and belong to the class of NST (no special type).

Response to reviewer comments:

Reviewer #1 (Remarks to the Author):

We thank Reviewer 1 for the very helpful critiques and the kind remarks.

Critiques verbatim:

In this manuscript the authors have overexpressed TRIM24 specifically in the mammary luminal epithelial cells through the use of MMTV-Cre. This resulted in tumors after a long latency with several histological subtypes. The subtypes somewhat resembled human MpBC and they illustrated survival differences in these tumors based on nuclear TRIM24 levels. Comparison of normal mammary to TRIM24 tumors identified a number of genetic pathways, several related to EMT and PI3K. Treating human PDX samples to degrade TRIM24 resulted in a reduction of viability and the authors suggest that there may be therapeutic benefits for TNBC patients.

Overall the manuscript is well written and readily understood with important findings for the role of TRIM24 in breast cancer.

There are major issues that may be relatively easily addressed with additional experiments. First, the experimental design starting with results shown in Figures 3 and 4 is highly flawed. Comparison of normal mammary gland to tumor yielded MANY genetic hits, but the problem is that they are comparing two states (normal and tumor) that are completely different. This was compounded by doing bulk sequencing, not single cell, so they are comparing a mammary gland composed of adipose, stroma and a minimal amount of mammary epithelium to a epithelial dense tumor.

Two better options would have been to;

1) Include TRIM24 transgenic mammary glands, both early (12 weeks) and from contralateral glands when the primary tumor was excised."

- When we did this comparison (Option 1), even uninvolved mammary glands – contralateral to the primary tumor site – showed marked differences by IHC for specific markers compared to normal mammary gland and were much more variable than the endpoint tumors. Our concern was that this comparison may be somewhat ambiguous, although interesting with regards to tumor evolution. We also followed Option 2, suggested by the reviewer, which offered an excellent means of comparison. Data gathered and changes as a result are now incorporated into the revised manuscript (see next point). Importantly, our results are greatly strengthened by these new data.

2) Compare the TRIM24 tumors to other characterized tumors. Microarray, RNAseq and scRNAseq data are all available for other mouse mammary tumor models. Indeed, the comparison to the MET tumors from Dr. Park's group begs the comparison to those tumors.

- The results of these comparisons to other (not over-expressing TRIM24) mouse mammary tumors (line 203-205) are now included in Figure 3B. The comparison to Dr. Park group's MET mouse model microarray (dataset GSE41748) and our RNAseq data are now shown as Suppl. Figure 3E (line 256-261).
- These comparisons show that TRIM24-driven carcinosarcomas are unique and segregated from non-TRIM24 driven tumors. One tumor expression profile of each type segregated

separately. Also, when the derived TRIM24 metaplastic score, calculated based on TRIM24 differential expression compared to MMTV-Cre mammary glands, is used to assess correlation with MET tumors from Dr. Park's group, the TRIM24-driven tumors are closely related to MET-driven tumors at a molecular level (line 256-261). We greatly appreciate the suggestions.

Setting aside the flaw in the experimental design that can be addressed with additional work, the manuscript was informative. Additional major concerns are as follows:

1) 8 week glands are shown in Figure 1 and it appears that branching is increased. Were the mice staged for estrus cycle? This should be done routinely.

2) Branching should be quantitated and compared with a statistical test at 8 weeks. This analysis should also be repeated for mice that have developed tumors – as well as for age matched controls.

- This is a great suggestion. We were able to quantify branching of glands, as described in the resubmitted manuscript (line 148-150). We used these data in updated Fig. 1B. These data fully support the qualitative assessment presented previously. The updated data were compared statistically, which supported the significant changes effected by TRIM24.
- For branching analyses, we dissected glands from parallel (cohort) littermate female mice of each genotype on the same day, harvested at the same time of day and at the same age to reduce effects of both estrous and diurnal cycling. We agree that staging mice by measured estrous cycle is a better plan that we will follow in future work. During the time of resubmission, our Covid-induced shutdown disrupted breeding/aging of appropriately matched mice in a cohort. We could not perform estrous cycle analyses of sufficient mice for branching analyses over the six-month time period of resubmission.

3) Given that only 40% of mice developed tumors, the KM plot showing controls with only 4 tumors is rather weak statistically. In addition, having only 12 mice develop tumors is exceedingly low, especially when the authors attempt to stratify histological subtypes. Additional tumors should be generated and characterized.

- Fortunately, we were able to increase the number of mice of Trim24^{COE} to 40. These additional mice took the survival curve out past 600 days. We were able to increase the number of MMTV controls for the survival curve to n=27. The overall penetrance is calculated as 46% with these additional mice (line 151-155). The timing of resubmission did not allow us to increase the lox-stop-flox numbers, as these were not being continuously aged with this cohort.

4) The point of Figure 1 Panel C is unclear – why is normal control mammary being compared to a tumor? Especially when the tumor histology is shown again in Figure 2.

- The figures have been rearranged to accommodate additional data. We elected to keep the control MMTV and Trim24^{COE} comparisons of TRIM24 expression as Fig. 1D to offer a good illustration of ductal morphology and how it is altered in the Trim24^{COE} mammary gland. Fig. 2 no longer has this repeated control.

5) A trained mouse pathologist should examine the tumors – not specialists in human breast cancer pathology. There are a number of talented mouse mammary pathologists that should be consulted and who would likely change the interpretation. While not a pathologist, the dark

arrow in Figure 2 Panel C image i DOES NOT denote stroma. Instead, this appears to be a tumor that is in the midst of being taken over by a spindleoid morphology, reminiscent of EMT. For suggestions of pathologists see the following:

- a. Robert Cardiff UC Davis (may be retired / retiring?)
- b. Tan Ince Weill Cornell
- c. Alexander Borowsky UC Davis

- Thanks for this suggestion, very much appreciated. We have a trained mouse pathologist at MD Anderson at each research location, Houston and Science Park, and they reviewed our slides. We added Drs. Mihai Gagea and Manu Sebastian to the list of authors to recognize their important contributions.
- The arrow, referenced in Fig. 2 C image I, was identified by our pathologist (and the reviewer), as follows: Fig. 2C) Comparison of H&E stained sections of i) murine *Trim24^{COE}* tumors and ii) human MpBC tumor at 100x magnification. Each tumor has a cohesive epithelial component (open arrow) and a discohesive spindle cell component (solid arrow). This was my error in the original manuscript and I thank you for pointing this out.

Reviewer #2 (Remarks to the Author):

We are grateful to Reviewer #2 for their time and their thoughtful critique.

Shah et al have created a transgenic mouse that conditionally overexpresses TRIM24 in mammary gland epithelial cells. TRIM24 is a chromatin reader protein that modulates p53 and metabolic pathways and that appears to promote cancer. Its expression correlates with poor survival of breast cancer patients. In this manuscript, the authors showed that this new line of mice is predisposed to breast cancer. The majority of tumors were carcinosarcomas, which based on the authors' characterization, serve as a model for multiplastic, triple negative breast cancer in humans. The authors then molecularly characterize both the mouse tumors and human patient tumors and identify c-MET and the PI3K/mTOR pathway as relevant targets that may be used to formulate therapeutic approaches. There are not any great mouse models for any kind of triple negative breast cancer (TNBC), the findings are mechanistic and advance the field, and the findings are of enough general interest to the cancer community to justify publication in a journal such as Nature Communications. That being said, the authors need to concentrate on strengthening the existing data and addressing a number of presentation issues.

1. The authors do not provide enough detail about the founder mice. Was there one line of founder mice or multiple? If multiple lines, were used, the authors should indicate where each was used. Was there any attempt to map the integration site(s)? If there is only one founder line, this may be an issue if the integration interrupted a critical gene or regulatory sequence.

- The one founder line is described in much more detail now, please see Methods (line 529-540) and description in Results (line 141-144). We mapped the integration site of the transgene by whole genome sequencing, discussed at lines 141-144 of the manuscript. The transgene integrates as a single copy into the q-arm of Chromosome 1 in a region with no identified genes, proximal to the telomere. Please see Suppl. Fig. S1E and Suppl. Table 3 of the resubmitted manuscript for the mapping diagram and data, as well as the whole genome sequencing data, respectively.

2. *MMTV-Cre can allow expression in tissues other than mammary epithelium. Were tumors observed anywhere other than the mammary gland? If so, it should be reported in a supplemental figure/table. Such results would not detract from what was learned about mammary/breast tumors, but should be reported.*

- Thank you for this suggestion. We have included Suppl. Table 4 to report the details of the aging cohort with regards to necropsied mice bearing tumors and any metastases. Interestingly, all mice in our cohort have mammary tumors and no primary tumors at other sites – even salivary glands. This important fact is now included in the manuscript (line 151-153).

3. *There appears to be a discrepancy - tumor penetrance was reported to be only 40%, but the decrease in survival indicates 75% of the mice died early compared to control mice. Does that mean there is a tumor-independent effect on survival? The authors should address this. Were all TRIM24-COE mice terminated after 400 days while the controls were left for 600 days?*

- We apologize for any confusion on this. The tumor penetrance of the TRIM24-COE mice is 46% with the added mice (now 40 COE mice) of the cohort taken out beyond 600 days (line 151). Eighteen TRIM24-COE mice of 40 total developed tumors. We added the calculation of penetrance to the Methods.
- The updated survival curve is Fig. 1C in the resubmitted manuscript. MMTV controls (n=27) were bred within the same time frame. Tumor-Free Survival was assessed for the aging cohort (line 151-155).

4. *Figs. 2E and S2B – The reviewer does not understand why the survival curve for patients with both nuclear and cytoplasmic staining was not shown and cannot be done without duplication, as the authors state. Isn't this a separate cohort of patients?*

- We are grateful for this query as our pathologists validated the patient samples for both nuclear and cytoplasmic scoring, as well as the specific subtype of MpBC present in each patient. This information is now included in lines 185-191 and Supplemental Table 5. The reviewer is correct that survival can indeed be assessed for four classes of patients based on TRIM24 expression nuclear/cytoplasmic: high/low. This is now included as new Fig. 2E and correlates directly with the scoring (Supplementary Table 5) and Fig. 2D image.

5. *One of the more significant issues is the authors' decision to combine transcriptional analysis of the carcinoma and the carcinosarcoma tumors (Fig. 3). Why not show both separately? The case has been made that these mouse carcinosarcoma tumors are similar to multiplastic patient tumors (MpBC), so why confuse the analysis by doing gene expression analysis on a mix of carcinoma and carcinosarcoma mouse tumors? The heterogeneity of the individual carcinoma tumors further argues against combining analysis of the different tumor types.*

- Thank you for pointing out this issue. The heatmap does show both carcinoma and carcinosarcoma profiles, now with added non-TRIM24 tumors as well as MMTV-cre controls (line 202-205). Our rationale for this comparison is that carcinomas may be molecularly closer to normal than carcinosarcoma. The heterogeneity of carcinoma tumors may be due to TRIM24 overexpression in mammary epithelia, although one carcinoma of each (TRIM24-expressing and TRIM24-nonexpressing) clustered separately. Carcinosarcomas - which are a majority of the TRIM24-driven tumors - are transcriptionally clustered much closer to

themselves and separated from other tumor types (carcinoma) and mammary glands. Pathway analyses were done with only carcinosarcoma data, which is now clarified in the text.

6. Another significant issue is the weak evidence for direct activation of c-MET by TRIM24. As presented, there is an increase in c-Met in the TRIM24 overexpressing mouse tumors and an unconvincing argument for direct regulation based on the ATAC-seq and ChIP data (addressed in more detail below). Perhaps the authors can knock down TRIM24 in the overexpressing cells or perform some other experiment to better link TRIM24 manipulation with c-MET expression levels.

- We agree with the reviewer that this evidence needed strengthening. Here, we addressed the issue of ChIP-PCR by performing PCR at multiple sites at cMET and updated Figure 5 B&C (line 309-312). We also followed the reviewer's suggestion to do knockdown of TRIM24 to support the direct regulation of Met transcription by TRIM24. This worked beautifully and is now included as Suppl. Fig. 4D with lines we created to express tet-regulated shControl and shTRIM24 (312-313).

The ChIP in Figure 5C is simply not convincing. First, the authors state that mononucleosomes were isolated. Yet the two primer sets are identified as "+307" and "+174", which are separated by less than one mononucleosome-length of DNA. So how could one reasonably expect different results for the two primer sets unless it were known that there were highly and invariantly positioned nucleosomes at that locus that caused the examined sequences to be incorporated into different nucleosomes? In addition, both primer sets are in regions of enhanced chromatin accessibility, so again, why would one necessarily believe that ChIP results would be different?

- Thank you for making this point and the primer sites were poorly identified. We took the opportunity in resubmission to perform several more ChIPs with more preparations and more primer sets (line 309-312). We performed ChIP-PCR for TRIM24 binding at cMET at various sites along cMET promoter and other regions and updated in Figure 5 B&C. Our prior conclusions were substantiated by these studies and strengthened. We find that a subset of open chromatin sites in TRIM24-COE cells, which are at the upstream regulatory region of Met, is TRIM24-bound. A downstream region of open chromatin is not TRIM24-dependent.

7. In Figure 6, the authors suddenly jump to PDX tumors as an experimental system and a PROTAC degrader as the tool for manipulation of TRIM24. This is problematic because of the lack of controls and the modest nature of the data.

(A) It's great that others have reported that the PROTAC degrades TRIM24 in a different cell type, but it really should be shown that TRIM24 levels were reduced/eliminated in the cells that were used for the experiments presented here. Westerns may not be possible, but presumably some form of IHC or ICC is possible.

- We felt the PDX studies were important in showing the relevance of our mouse model to human disease, as well as an opportunity to assess TRIM24 expression and response to the PROTAC outside of established cell lines. This is the first time, to our knowledge, that the TRIM24 PROTAC was tested in a solid tumor setting (now noted in the text).
- The opportunity to undertake a study of a TRIM24 PROTAC with MpBC PDX and additional non-MpBC TNBC PDX allowed this comparison, but we can make no claims of established

therapeutic indicators, based on these data, especially given the few MpBC PDX that exist. Any suggestion that these data indicated MpBC-specific therapeutic potential has been removed from the text. Since we added more TRIM24 IHC of MpBC and non-MpBC TNBC PDX samples and found that all express TRIM24, with all MpBC and 2 of 4 non-MpBC TNBC PDX responding with loss of viability to the TRIM24 PROTAC, the modification of the text to indicate a therapeutic potential in the future for TRIM24-expressing TNBC patients (not MpBC-specific) is more justifiable.

- Showing the PROTAC works as expected in our hands is a very important point. We were able to do a time course treatment of murine Trim24^{COE} tumor and control cells with the TRIM24 PROTAC to show effective degradation of TRIM24 with dTRIM24 PROTAC and none with eTRIM24 control (line 408-410). This is now included in the resubmitted manuscript as Suppl. Fig. 5C. Unfortunately, our collaborator, who developed the PDX models, does not maintain any numbers that are sufficient for in vivo treatment with any drugs, in order to compare treated versus untreated tumors, which would have been ideal. Additionally, over a time course of resubmission, this would not have been doable.

(B) Even assuming the controls are appropriate, the data presented are weak. 4/4 MpBC PDXs showed an effect, but one was marginal. 2 of 3 TNBC PDX samples showed no effect, but one showed a significant effect. Perhaps the authors can buttress the PDX data with inhibitors or by including more samples or by some other means.

- We agree with the reviewer and it would be great to have more MpBC PDX models. However, please note that available MpBC PDX are few, as this is a rare subtype of TNBC. Another TNBC (non-MpBC) PDX was included in an additional screen and shown to be responsive to dTRIM24 PROTAC. There are no other MpBC TNBC PDX that are available to us. The assays were independently conducted a sufficient number of times with enough samples to assess statistical significance.
- The text has been rewritten in light of the updated data. We now stress that we find TRIM24 over expression in multiple subtypes of TNBC and MpBC (Supplemental Fig. 5 and Fig. 7) (line 393-396) and further reference previous reports that TRIM24 is also over expressed in other subtypes of breast cancer. The responsiveness of four of four MpBC PDX and two of four other TNBC PDX to the dTRIM24 PROTAC indicate some potential for pursuing TRIM24 as a therapeutic target in TNBC.

8. There is no indication that any of the -omics data have been placed in an appropriate repository.

- RNA-Seq data from both mouse and human MpBC are deposited on GEO, as referenced with Ascension numbers in the Methods section (RNA-Seq and TRIM24 metaplastic signature). ATAC-Seq data are also submitted on GEO and can be found in Methods under ATAC-Seq analysis. WGS data are not supported on the GEO platform; hence, we provided Suppl. Table 3 with all the discordant locations. These references are also included in a separate section, Data Availability (line 740-748).

Issues with presentation, in order of appearance:

9. The last sentence of the abstract is awkwardly worded. Please clarify.

- We completely agree. It has been removed.

10. *None of the Supplemental tables are easily identified. Each table lacks a label indicating the table number and a title indicating what content is being shown. Please address.*

- We will make sure that this doesn't happen during PDF conversion, when we submit it to the portal. The table number and titles are also present in the manuscript text right after the supplementary figures (line 1118). We apologize for any inconvenience.

11. *lines 131-132 "...injected into the pronuclei of FVB/NCr (Charles River Laboratories) zygotes ..." and lines 138-139, "...by breeding MMTV-Cre mice with FVB...". The authors do not describe the strain nor do they explain why it was chosen.*

- We apologize for not mentioning this in text. We have updated the Methods section with the rationale behind the choice of FVB mice for TRIM24 overexpression (line 515-516). The FVB strain is known to be predisposed to mammary tumor development, similar to the Balb-C strain.

12. *lines 139-140 "Virgin female mice were aged until there was evidence of a palpable tumor." Where were tumors located? Were they all mammary tumors? If not, what was observed and which tumors were used for subsequent analyses?*

- Thank you and Reviewer 1 for pointing out this oversight. The new Suppl. Table 4 describes all the mice with tumors and sites of metastasis. We have not observed primary tumors at any sites other than mammary glands (added to main text). MMTV has been found previously to drive tumors of the salivary gland, sometimes dependent on the transgene, but not with TRIM24.

13. *lines 194-199 need to be corrected for grammar.*

- We corrected our mistake and added the missing text back (line 587-590).

14. *Fig. 1D and Supp Fig. 1E should be combined and presented together. It's an IHC staining – why does the reader have to go to the supplement to see the control so that the conclusion can be evaluated?*

- We have combined them in Figure 1E as per your suggestion. We appreciate the suggestion as this does make more sense.

15. *The negative results in Fig. S2A require images of the positive control run with these tumors. Similarly, Fig. S5A requires a negative staining control.*

- Definitely. We have updated Fig S2A and included Positive Control staining. For Fig S5A, now Fig S5B, we have included positive and negative staining for human mitochondria using human intestine samples.

16. *Fig. 3B/D lack a scale relating the colors to quantification.*

- We have addressed this in text and figure legend.

17. There are labeling issues in Figs. 3 and 6. Fig. 3 shows a transition from referring to the mouse carcinosarcoma tumors to referring to them as metaplastic tumors, with the abbreviation of “M”. While by definition this is technically correct, it causes confusion with data from metaplastic breast cancer patients. In Figs. 4 and 5 there is a resumption of use of “carcinosarcoma” for labeling. To be consistent and to minimize confusion, the reviewer recommends that “carcinosarcoma” be used in Fig. 3.

- Thank you for making this point, as one loses sight of such. To distinguish the mouse mammary carcinomas (C) from the mouse mammary carcinosarcomas, or metaplastic mammary tumors, we used (M). Throughout the text we now use the term metaplastic carcinosarcoma, when referring to the murine carcinosarcoma tumors, and MpBC, when referring to human metaplastic TNBC. We also make the point to use MpBC TNBC and non-MpBC TNBC, when appropriate, to underscore that MpBC refers to a rare subset of TNBC patients.

In Fig. 6, the authors again refer to the mouse carcinosarcoma tumors as Metaplastic (M). But then in Fig. 6F, “M” stands for “mesenchymal”. Again, it might decrease confusion if “carcinosarcoma” were used to refer to the mouse mammary tumors, “M” were used for the human tumors, and a different abbreviation were used for “mesenchymal”.

- The abbreviations and different “M”s do get confusing. Hopefully, our changes will clarify these references. We addressed the potential confusion introduced by the reference to the mesenchymal subtype by replacing “M” for mesenchymal to “M*⁺”. This is underscored in the figure legend of Figure 6.

18. line 516-517 reads “..., which was significant only for MpBC (Fig. 6C).” This is open-ended, and should be reported as “significant for MpBC but not non-metaplastic TNBC”, or similar.

- We addressed this, per your suggestion, and further took care to delineate MpBC TNBC versus non-MpBC TNBC (line 366).

19. line 521 introduces “RCB-II and RCB-III outcomes”. “RCB” is not defined, the “outcomes of II and III” are not defined, and the significance of these outcomes is not defined.

- We have addressed this oversight and provided the definition of RCB and more detailed discussion (line 370-372).

20. Would the graph in Fig. 6A be better presented with TNBC and MpBC as labels rather than No and Yes? Or perhaps consider putting a key within Figs. 6C-F to explain that the red and black dots are MpBC and TNBC patients.

- We have addressed this in the figure legend, as well as showing what the red and black dots are as a key in the graphs. Thanks for the suggestion.

21. line 549. What are MOLM-13 cells and why are they relevant?

- We apologize for the confusion. MOLM13 cells are acute monocyte leukemia (AML)-derived cells. We now include that in the text and the fact that these leukemia cells were responsive to the PROTAC, whereas no 2D-attached cell lines – including breast cancer-derived – were

responsive (line 404-406).

Reviewer #3 (Remarks to the Author):

We greatly appreciate the time and effort of Reviewer 3. The suggestions made were helpful in improving our manuscript. Please see critiques, addressed below:

The manuscript is based on preclinical data showing that transgenic mice over-expressing Trim24 develop mammary tumours resemble to metaplastic carcinomas. Global and single-cell tumor profiling of these murine tumors revealed Met as a direct oncogenic target of TRIM24, leading to aberrant PI3K/mTOR activation. Pharmacological inhibition of these pathways in primary Trim24 tumor cells and TRIM24-PROTAC treatment of MpBC PDX tumorspheres revealed the therapeutic potential of targeting TRIM24.

The findings are new and present some clear interest from a molecular and therapeutic point of view for metaplastic breast cancers.

The major limitations are that the human relevance is not sufficiently explored and in vivo preclinical experiments are lacking to validate the therapeutic potential of TRIM24 targeting.

1. Figure 2A shows the frequency of the different types of sarcoma developed in transgenic mice. How many mice and how many tumours have been analysed in total?

- Thank you for this suggestion. We added more mice to the survival curve of Fig. 1C (40 Trim24COE mice) and now provided a Suppl. Table 4 with the details of tumor location and number per mouse, as well as any metastases with the locations. Details are now added to the main text as well (line 151-155, 165).

2. The clinical and histological characteristics of the 46 human metaplastic breast cancers used to validate TRIM24 expression are not fully described. Which types of metaplastic breast cancers are analysed (chondroid, spindle, squamous?) Moreover only 10% seem to over-express TIM24. This is an important point to validate Trim24 overexpression in MBC.

- Thank you for this suggestion, which added considerably to our comparisons. We have now provided these details of types of MpBC in Suppl. Table 5. We have revised the graph of Overall Survival to be more clear on nuclear and cytoplasmic expression of TRIM24, included as Fig 2E, which clearly show the impact of nuclear TRIM24 and its correlation with worse patient survival (line 185-190).
- It is now possible to see in Suppl. Table 5 that 37% of the 46 MpBC patients assessed had high nuclear TRIM24 expression by IHC and 43% had high cytoplasmic TRIM24 expression. We have also added details with regards to subtype of MpBC and TRIM24 expression to the main text (line 185-190). These were important additions.

3. Additional cohorts of human TNBC and MBC should be analysed to provide evidence of TRIM24 over-expression is a characteristic of metaplastic breast cancer. In silico analysis of public datasets such as cBioportal should also be analysed (the TCGA dataset in cBioportal includes a group of metaplastic breast cancer).

- Thank you for this suggestion. We went to cBioportal to analyze MpBC data, but unfortunately there are no case data of MpBC available to associate with TRIM24 (see reviewer-only figure below). We compared additional human MpBC patient data that were publicly available (GSE57544) by calculation and comparison of the TRIM24 Metaplastic signature scores (Suppl. Fig. 3D) (line 250-255). We also compared the RNA-seq data of our TRIM24 metaplastic carcinosarcoma tumors with another MpBC human patient cohort by GSEA enrichment plot and found significant correlations by both approaches (Suppl. Fig. 3F).

4. What drives TRIM24 over-expression in human breast cancer? An analysis in cBioportal could be performed to analyse TRIM24 expression and gene amplification/gains in different breast cancer datasets. If TRIM24 over-expression is driven by gene amplification, is there an enrichment of TRIM24 amplifications in metaplastic breast cancers?

- Thank you for asking this important question. We are currently validating a CRISPR screen that will allow us to address the regulators of TRIM24 comprehensively. Knowing exactly what drives TRIM24 over expression in human breast cancer is beyond the scope of this manuscript, but we have taken the reviewer’s suggestion and determined it to be at the level of gene amplification. Since the MpBC cohort is very small and there is not enough information on cBioportal regarding amplification and mRNA expression for TRIM24, we decided to look in general using all BC patient data. Yes, there is a positive correlation between TRIM24 amplification and mRNA expression as shown in the reviewer-only figure below.

5. The analysis of TRIM24 expression in PDX models is done without a statistical analysis: how many TNBC PDX were analysed? Only 1 is shown in figure 7A. Ideally different groups of TNBC should be analysed (basal-like, LAR, mesenchymal tumors , etc.).

- Thank you for your question, and we apologize for not including this information earlier. We have analyzed 4 MpBC and 4 TNBC PDXs. We have updated Fig 7A and Fig S5B with IHC for TRIM24 and human mitochondria stain for all PDXs analyzed for this study. We have included Vanderbilt subtype information for all PDXs in Fig S5A, which include BL2, mesenchymal, LAR, and BL1 subtypes

6. The data presented in figure 7B are not sufficient to conclude that TRIM24 is a therapeutic target. What is the expression of TRIM24 in the 3 TNBC PDX? are they all negative? In vivo studies should be performed in different PDX models of metaplastic breast cancer.

- This is a good point and we now suggest TRIM24 as a potential therapeutic target for TRIM24-expressing TNBC patients. We completely agree to say it is a therapeutic target for MpBC specifically is not supported.
- We have included TRIM24 IHC for all PDXs including 4 non-MpBC TNBC PDXs in Fig7A. All the TNBC PDX tumors, used in our PROTAC analyses, expressed TRIM24 at higher levels than another TNBC PDX, which was not used in our PROTAC studies (Fig. S5C). It is too premature by far to state that TRIM24 is a therapeutic target for MpBC.
- Additionally, we have included a western blot analysis of TRIM24COE tumor primary cells treated with DMSO, eTRIM24 and dTRIM24 and have shown that dTRIM24 reduces TRIM24 protein level as expected (line 408-410).
- Our collaborator, who developed the PDX models, does not maintain any numbers that are sufficient for in vivo treatment with any drugs, in order to compare treated versus untreated tumors, which would have been ideal. MpBC, as a rare subtype of TNBC, has few PDX models and the total number we have access to is included and analyzed in this manuscript. We would indeed like to have several more to compare.

7. *Figure 6F does not show a clear enrichment of the TRIM24 signature in the MSL subtype. Moreover, in the M subtype there are many tumors that are not metaplastic breast cancers and belong to the class of NST (no special type).*

- We apologize for this impression. We now clearly state in the text that the Metaplastic signature tumors are found in M, MSL and IM subtypes (line 376). We incorrectly defined UNS as unspecified, and this may have led to the reviewer thinking that class was “no special type”. It is UNS=Unstable. We have altered the text to better represent the data and the subtype definition. Thank you for pointing this out.

REVIEWERS' COMMENTS

Reviewer #1 (Remarks to the Author):

I appreciate how much work was completed in response to the review and hope that the authors agree that the manuscript is now markedly better.

I have no additional concerns and encourage the editors to accept the present version.

Reviewer #2 (Remarks to the Author):

I have finally reviewed the revised manuscript by Shah et al. Although my original review (reviewer 2) was lengthy, the vast majority of the issues related to points that needed clarification or further development or to presentation issues. All of these have been adequately resolved; potential data interpretation issues were largely explained. Requests for data have been added without raising any subsequent issues. The only complaint I have is that the requested TRIM24 knockdown experiment (Supp Fig. 4D) is only validated by qPCR and not by Western or other analysis of protein levels. However, I cannot see holding up the manuscript for this point.

I recommend acceptance of the manuscript.

Reviewer #3 (Remarks to the Author):

1) Authors did not adequately respond to my comment n°3 "additional cohorts of human TNBC and MBC should be analyzed to provide evidence that TRIM24 over-expression is characteristics of metaplastic breast cancer."

This point is important as it would validate preclinical finding on murine tumors.

In the TCGA dataset (960 samples) on cBioportal a subset of breast ductal carcinoma and not metaplastic breast cancer over-express the TRIM24 gene (see graph in the review document).

2) in vivo experiments should be performed especially if the objective is to demonstrate that TRIM24 positive tumors could be treated with c-MET or PIK3 inhibitors

Response to Reviewers' Comments

We are very grateful to the reviewers' time and commitment to offer helpful suggestions to improve our manuscript. Please see below our response to this round of critiques:

REVIEWERS' COMMENTS

Reviewer #1 (Remarks to the Author):

I appreciate how much work was completed in response to the review and hope that the authors agree that the manuscript is now markedly better.

I have no additional concerns and encourage the editors to accept the present version.

Response:

Thank you! We do appreciate how much better the current version of the manuscript is.

Reviewer #2 (Remarks to the Author):

I have finally reviewed the revised manuscript by Shah et al. Although my original review (reviewer 2) was lengthy, the vast majority of the issues related to points that needed clarification or further development or to presentation issues. All of these have been adequately resolved; potential data interpretation issues were largely explained. Requests for data have been added without raising any subsequent issues. The only complaint I have is that the requested TRIM24 knockdown experiment (Supp Fig. 4D) is only validated by qPCR and not by Western or other analysis of protein levels. However, I cannot see holding up the manuscript for this point.

I recommend acceptance of the manuscript.

Response:

Thank you! We were torn about presenting the shRNA validation of the TRIM24 knockdown by western or qRT-PCR. Since Met is regulated by TRIM24 at the transcript level, not protein, and the shRNA for Trim24 targets RNA, we focused on RNA expression analysis. Thank you for your appreciation of the major changes we made in resubmission.

Reviewer #3 (Remarks to the Author):

1) Authors did not adequately respond to my comment n°3 "additional cohorts of human TNBC and MBC should be analyzed to provide evidence that TRIM24 over-expression is characteristics of metaplastic breast cancer."

This point is important as it would validate preclinical finding on murine tumors. In the TCGA dataset (960 samples) on cBioportal a subset of breast ductal carcinoma and not metaplastic breast cancer over-express the TRIM24 gene (see graph in the review document).

Response:

Thank you for further clarification. We previously included discussion that TRIM24 was expressed in multiple cancers, as one factor in our decision to create a mouse model of conditional Trim24 over expression. We did not mean to imply that only metaplastic TNBC (MpBC) over express TRIM24. We have now added additional discussion that TRIM24 RNA is over expressed in multiple human cancers and that, as TRIM24 is regulated post-transcriptionally, we developed a molecular, multi-gene signature of TRIM24 impact on gene expression. We then used this gene signature to determine how similar MpBC versus other subsets of TNBC is to the Trim24 metaplastic carcinosarcomas that formed as a result of Trim24 over expression.

We have now added more text to describe in the manuscript: i) how TRIM24 is regulated and ii) why we used a gene signature of TRIM24-regulated gene targets to determine the impact of deregulated gene expression in tumors. Further, we added why this approach was limited to analysis of the MpBC and non-MpBC TNBC patient samples and how this signature may be applied more universally to other subtypes of breast cancer or other human considers, although batch- and tissue-specific variables may confound these analyses. Hopefully, our readers will understand both why a gene signature is more accurate in assessment of TRIM24 impact and how that signature may be used in other analyses.

2) in vivo experiments should be performed especially if the objective is to demonstrate that TRIM24 positive tumors could be treated with c-MET or PIK3 inhibitors

Response:

This is exactly what needs to be done in future preclinical studies of both inhibitors of c-Met and PI3K, as well as any TRIM24-targeted PROTAC. However, our collaborator in development of the MpBC PDX models does not maintain these PDX in sufficient numbers, so breeding and passaging is needed over considerable time, and further stated that these preclinical studies are likely to take 2-3 years. We consider these as important next steps (discussion to point this out added to manuscript) but beyond the scope of the current manuscript.